# Restyling Unsupervised Concept Based Interpretable Networks with Generative Models

**Jayneel Parekh**[⋆,1,2]    **Quentin Bouniot**[⋆,2,3,4,5]    **Pavlo Mozharovskyi**[2]
**Alasdair Newson**[1]    **Florence d'Alché-Buc**[2]
[1]ISIR, Sorbonne Université,    [2]LTCI, Télécom Paris, Institut Polytechnique de Paris, France,
[3]Technical University of Munich,    [4]Helmholtz Munich,
[5]Munich Center for Machine Learning (MCML)

## Abstract

Developing inherently interpretable models for prediction has gained prominence in recent years. A subclass of these models, wherein the interpretable network relies on learning high-level concepts, are valued because of closeness of concept representations to human communication. However, the visualization and understanding of the learnt unsupervised dictionary of concepts encounters major limitations, especially for large-scale images. We propose here a novel method that relies on mapping the concept features to the latent space of a pretrained generative model. The use of a generative model enables high quality visualization, and lays out an intuitive and interactive procedure for better interpretation of the learnt concepts by imputing concept activations and visualizing generated modifications. Furthermore, leveraging pretrained generative models has the additional advantage of making the training of the system more efficient. We quantitatively ascertain the efficacy of our method in terms of accuracy of the interpretable prediction network, fidelity of reconstruction, as well as faithfulness and consistency of learnt concepts. The experiments are conducted on multiple image recognition benchmarks for large-scale images. Project page available at https://jayneelparekh.github.io/VisCoIN_project_page/

## 1 Introduction

Deep neural networks (DNNs) learn complex patterns from data to make predictions or decisions without being explicitly programmed how to perform the task. *Interpreting* decisions of DNNs, *i.e.* being able to obtain human-understandable insights about their decisions, is a difficult task (Beaudouin et al., 2020; Arrieta et al., 2020; Montavon et al., 2019). This lack of transparency impacts their trustworthiness (Rudin et al., 2022) and hinders their democratization for critical applications such as assisting medical diagnosis or autonomous driving. Two different paths have been explored in order to interpret DNNs outputs. The simplest approach for practitioners is to provide interpretations *post-hoc*, *i.e.* by analysing the so-called *black-box* model *after* training (Baehrens et al., 2010; Ribeiro et al., 2016; Lundberg & Lee, 2017; Selvaraju et al., 2017). However, *post-hoc* methods have been criticized for their high computational costs and a lack of robustness and faithfulness of interpretations (Yosinski et al., 2015; Kindermans et al., 2019; Alvarez-Melis & Jaakkola, 2018b). On the other hand, one preferred way to obtain more meaningful interpretations is to use *interpretable by-design* approaches (Al-Shedivat et al., 2017; Adel et al., 2018; Böhle et al., 2022; Gautam et al., 2022), that aim to integrate the interpretability constraint into the learning process, while maintaining state-of-the-art performance.

*Concept-based Interpretable Networks (CoINs)* are a recent subcategory of these inherently interpretable prediction models, that learn a dictionary of *high-level concepts* for prediction. The concept representation is either learnt in a supervised way using ground-truth concepts annotations (Koh et al., 2020; Sarkar et al., 2022), or in an unsupervised fashion by enforcing properties through carefully designed loss functions (Alvarez-Melis & Jaakkola, 2018a; Parekh et al., 2021; Sarkar et al., 2022). The output of the model can be interpreted by looking at the *activations* of each concept

---

⋆ Equal contribution

and how they are combined to obtain the final prediction. When working in the unsupervised setting, learnt concepts have to additionally be interpreted, usually through *visualization* (Parekh et al., 2021; Sarkar et al., 2022). *Concept-based interpretations* have gained prominence as an alternative to popular feature-wise saliency maps (Springenberg et al., 2014; Ribeiro et al., 2016; Lundberg & Lee, 2017; Selvaraju et al., 2017) for two main reasons: (1) their ability to provide interpretations closer to human reasoning and communication (Yeh et al., 2019), and (2) in specific case of visual modalities, their ability to more effectively highlight *which* features are important for a model and not just *where* in the input image they focus on (Colin et al., 2022). However, the underlying concepts in current unsupervised CoINs are understood through a *separate visualization pipeline*, by finding inputs that highly activate a given concept, either from natural images in the available dataset (Alvarez-Melis & Jaakkola, 2018a; Sarkar et al., 2022), or from virtual images by solving an optimization problem in the input space that maximally activates the concept (Mahendran & Vedaldi, 2016; Parekh et al., 2021). For large-scale images, concepts generally activate for local pattern information (color, texture, shape etc.) and these visualization approaches face major limitations in highlighting this information to a user. Simply visualizing the most activating samples does not highlight the specific feature a concept activates for. Visualizing using an activation maximization procedure leads to the generation of repeated patterns linked to the underlying concept in the image, but are hard for a user to discern any human-interpretable signal. For example, in Fig. 1, it can be hard to identify that the concept activates for "Yellow-colored head" from the activation maximization ("FLINT visualization"). Furthermore, previous CoIN systems fail to include the visualization process in their quantitative evaluation of concepts and their use-case for interpretation.

We thus propose a novel set of specifications for the concepts to be learnt: additionally to *fidelity to output* (predictive capability from the concepts), *fidelity to input* (encoding input relevant information in concepts) and *sparsity* (a few concepts activated simultaneously), we also promote the *viewability* of concepts during training. This viewability is now defined as the ability of the system to reconstruct *high-quality* images from the learnt concepts, by leveraging a *pretrained generative model*. In order to obtain this viewability property, we propose to learn a *concept translator*, *i.e.*, a *mapping* from the *concept representation space* to the latent space of the generative model. Learning the concept translator along with the other parameters of this novel CoIN system helps to improve the quality of the concepts. Finally, interpretation of concepts is obtained through *translation* to the generative model, allowing for a more granular and interactive process. Our contributions are:

(i) We propose *Visualizable CoIN (VisCoIN)*, a novel architecture for unsupervised training of CoINs relying on a *concept translator* module that maps concept vectors to the latent space of a pretrained generative model.

(ii) We introduce a new property for unsupervised CoIN systems, related to *viewability*. This property is imposed during the training of the system by enforcing *perceptual similarity* of the reconstruction, in addition to other constraints, and made possible by the use of a generative model.

(iii) We define a novel concept interpretation pipeline based on the concept translator and the associated generative model that allows to both obtain a high-quality and more comprehensive visualization of each concept.

(iv) We introduce new metrics in the context of unsupervised CoINs to evaluate the quality of concepts learnt, from the point of view of *visualization* and its usage for interpretation. We then quantitatively and qualitatively evaluate our proposed method on three different large-scale image datasets, spanning multiple settings.

## 2 RELATED WORKS

**Interpretable predictive models** In the context of deep learning architectures, a host of early approaches studying interpretability tackled the post-hoc interpretation problem (Simonyan et al., 2013; Springenberg et al., 2014; Ribeiro et al., 2016; Lundberg & Lee, 2017; Sundararajan et al., 2017; Chen et al., 2018). However, previous works such as those of Al-Shedivat et al. (2017); Li et al. (2018); Alvarez-Melis & Jaakkola (2018a) have contributed to surge of developing predictive models that are also interpretable *by-design* (Yoon et al., 2018; Agarwal et al., 2020; Zhang et al., 2018a; Lee et al., 2019; Gautam et al., 2022). The earlier systems, however, trained the complete

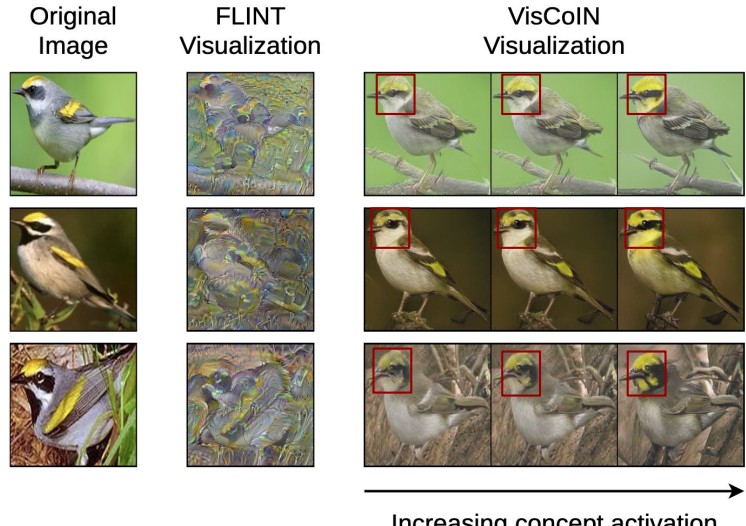

Figure 1: Comparison of the generated images obtain for the same learnt concept (*"Yellow-colored head"*) using FLINT visualization (Parekh et al., 2021) and our proposed VisCoIN visualization (in red boxes). Using our *concept translator*, that maps concept representation space to the latent space of a generative model, we can visualize each concept at different activation values, allowing for more granular and interactive interpretation. Visual modifications manually indicated by red boxes.

model from scratch. Recent approaches reflect a growing interest in building interpretable models on top of pretrained models (Koh et al., 2020; Angelov et al., 2023). Our approach falls in the latter category, wherein we learn an interpretable predictive model on top of a pretrained backbone.

**Generative models for interpretations** One of the earliest applications of generative models for interpretability was by Nguyen et al. (2016) to synthesize image for visualizing neurons in a network using GANs. More recently, a variety of methods have employed generative models for post-hoc counterfactual interpretations (Zemni et al., 2022; Lang et al., 2021; Farid et al., 2023; Ghandehar- ioun et al., 2021). Their central theme revolves around the idea of embedding any given input to the latent space of a generative model and finding meaningful perturbations in the latent space that affect the given predictor's output the most. One recent work (Ismail et al., 2023) also included the task of learning supervised concepts within generative models, to be able to interpret and steer their latent spaces. Our aimed use-case of generative models differs in a major way from these methods, because we wish to use it in order to learn and visualize an explicit dictionary of interpretable con- cept representation, simultaneously used in a predictive model.

**Concept-based interpretability** Providing interpretations via representations of high-level *con- cepts* has gained significant prominence recently. Similar to the overall literature, one set of concept- based methods have focused on post-hoc interpretation (Ghorbani et al., 2019; Yeh et al., 2019; Lang et al., 2021; Achtibat et al., 2022; Fel et al., 2023), with most based on the notion of concept activation vectors (Kim et al., 2017). The other type of methods tackle the by-design/ante-hoc inter- pretation problem by learning concepts (Alvarez-Melis & Jaakkola, 2018a; Koh et al., 2020; Parekh et al., 2021; Sarkar et al., 2022; Sawada & Nakamura, 2022; Sheth & Ebrahimi Kahou, 2024) ab- breviated as CoIN systems in Section 1. We cover these methods in more detail in Section 3.1 with particular focus on networks based on learning completely unsupervised concepts (Alvarez-Melis & Jaakkola, 2018a; Parekh et al., 2021; Sarkar et al., 2022; Garg et al., 2024), a key starting point of our approach. Recent variants of concept bottleneck models using language models (Oikarinen et al., 2023; Panousis et al., 2024) and why concept visualization is still useful despite their ability to obtain automated text descriptions, is discussed in detail in Appendix A.

## 3 APPROACH

### 3.1 BACKGROUND

In this part, we provide an overview of a concept based interpretable network (CoIN). Our focus in this paper is on CoIN systems that learn an unsupervised dictionary of concepts.

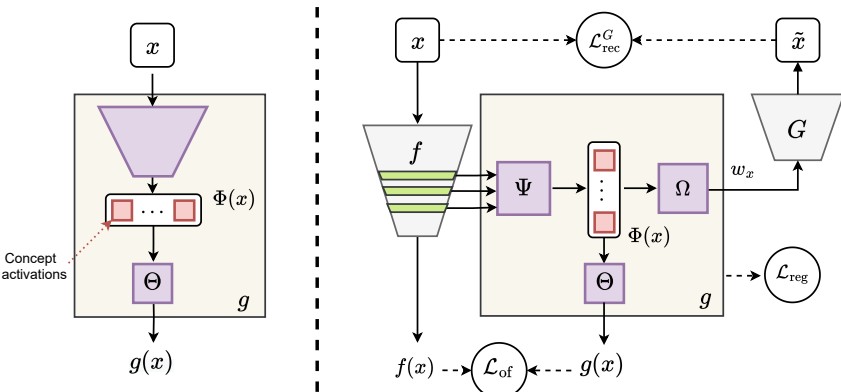

Figure 2: **Left:** Overview of a standard CoIN system $g$, that makes prediction $g(x)$ from extracted concepts $\Phi(x)$. **Right:** Design of our unsupervised concept-based interpretable network *VisCoIN* leveraging a pretrained generative model $G$ for visualization, and a pretrained classifier $f$. Purple blocks denote trainable subnetworks.

**Concept-based interpretable networks** We denote a training set for a supervised image classification task as $\mathcal{S} = \{(x_i, y_i)\}_{i=1}^N$. Each input image $x \in \mathcal{X} \subset \mathbb{R}^n$ is associated with a class label $y \in \mathcal{Y}$, a one-hot vector of size number of classes $C$. The *by-design interpretable classification network* based on learning concept representation is denoted by $g : \mathcal{X} \to \mathcal{Y}$. In the standard setup for concept-based prediction models (supervised or unsupervised), given an input $x$, the computation of $g(x)$ is broken down into two parts. There is first a *concept extraction representation* $\Phi$, and then a subnetwork $\Theta$ that computes the final prediction using *concept activations* $\Phi(x)$, such that $g(x) = \Theta \circ \Phi(x)$ (Fig. 2, left). *Supervised* concept-based networks use *sample-wise ground-truth concept annotations* to train $\Phi$. For instance, concept bottleneck models (CBM) (Koh et al., 2020) use a human-annotated binary concept vector denoting presence/absence of each concept. Although language model based CBMs do not require manual annotation of concepts, they also follow a very similar paradigm, e.g. by using CLIP "image-concept description" similarities as proxy annotations (Oikarinen et al., 2023; Yang et al., 2023a; Panousis et al., 2024). The core of *unsupervised* concept-based methods instead lies in learning $\Phi$ by imposing *loss functions*. These loss functions are typically selected to encourage a certain set of properties that shape $\Phi$ simultaneously for both interpretation and prediction. We list the properties below:

1. **Fidelity to output**: This requires $\Phi(x)$ to model the output space via the $\Theta$ function, either by predicting ground-truth label $y$ (Alvarez-Melis & Jaakkola, 2018a) or the classification output $f(x)$ (Parekh et al., 2021; Sarkar et al., 2022). It trains $\Phi(x)$ for the prediction task and, during the interpretation phase, helps in identifying important concepts for prediction.

2. **Fidelity to input**: This requires $\Phi(x)$ to reconstruct the input $x$ via a *decoder* function. This property is considered important to encode semantically meaningful input features in the concept representation. All the previous methods (Alvarez-Melis & Jaakkola, 2018a; Parekh et al., 2021; Sarkar et al., 2022) rely on this loss and employ standard *non-generative* decoders for pixel-wise reconstruction to learn the concept dictionary $\Phi$. Note that CBMs don't adhere to this property, leading to major design and training differences, as they don't include a decoder compared to unsupervised CoINs, that remain the focus of this work.

3. **Sparsity of activations**: This requires concept activations $\Phi(x)$ to be sparse for any $x$. It reinforces the high-level nature of $\Phi$ and limits the number of important concepts for prediction, thus enhancing interpretability and reducing the visualization overhead for users.

**Limitations with concept visualization in previous unsupervised CoINs** A common trait among prior CoINs learning unsupervised concepts (Alvarez-Melis & Jaakkola, 2018a; Parekh et al., 2021; Sarkar et al., 2022; Garg et al., 2024) is the deployment of a decoder to reconstruct input $x$ from $\Phi(x)$. Unlike supervised methods, they do not have access to any concept labels and thus need an additional visualization pipeline to understand the information encoded by each concept. However, their visualization pipeline does not utilize the decoder, but instead relies on proxy methods to probe the concept activation. Typically, it consists of finding an input that highly activates a concept, either by selecting from the training data (Alvarez-Melis & Jaakkola, 2018a; Sarkar et al., 2022) or

via input optimization (Parekh et al., 2021). In the former case, simply visualizing the set of most activating training samples lacks granularity to highlight the features encoded by the concept. Using input optimization, while relatively more insightful, is still difficult for a user to understand as the optimized images are often unnatural. Moreover, these issues exacerbate for large-scale images, as seen in Fig. 1. A natural strategy to overcome these limitations is to enable direct control of a concept's activation and visualizing its effect on the input. Since the decoder defines the relationship between concept activations and input samples, a *generative model* is a perfect candidate for a decoder to unlock this ability, in contrast to standard decoders used previously. While Garg et al. (2024) includes a GAN as a decoder model, it is learned simultaneously. This makes the overall training challenging, since GANs are notoriously difficult to train. Furthermore, they do not leverage the GAN for visualizing the concepts, only relying on maximum activating samples (MAS) for visualization. The GAN is used as a decoder with higher expressivity, to improve accuracy.

In the next part, we describe the architecture behind our *by-design interpretable network* $g$, that additionally includes a *concept translator* module $\Omega$, to map concept features $\Phi(x)$ to the latent space of a pretrained generative model $G$. This collectively defines our *VisCoIN* method.

## 3.2 INTERPRETABLE PREDICTION NETWORK DESIGN

The complete design of VisCoIN is illustrated in Fig. 2 (right). We assume a fixed pretrained network for classification $f$ and a fixed pretrained generator $G$, for generation on the input dataset respectively. We use these two networks to guide our design and learning of $g$ and its concept extraction function $\Phi$. We first discuss modelling of $g$ by describing its constituents, $\Phi$ and $\Theta$.

**Interpretable network design**   The *dictionary* $\Phi$ consists of $K$ *concept functions* $\phi_1, \ldots, \phi_K$. Given an input $x$, each concept activation $\phi_k(x)$ is represented by a small convolutional feature map with non-negative activation. Thus $\Phi(x) = [\phi_1(x), ..., \phi_K(x)] \in \mathbb{R}_+^{K \times b}$, where $b$ is the total number of elements in each feature map. We model computation of concept activations $\Phi(x)$ using the pretrained classification network $f$ and learn a relatively lightweight network $\Psi$ on top of its selected hidden layers denoted as $f_{\mathcal{I}}(x)$, *i.e.* $\Phi(x) = \Psi \circ f_{\mathcal{I}}(x)$. $\Theta$ is designed to simply pool the feature maps to obtain a single concept activation of size $K$ and make the final prediction by passing it through a linear layer followed by softmax, *i.e.* $g(x) = \Theta(\Phi(x)) = \text{softmax}(\Theta_W^T \text{pool}(\Phi(x)))$, where $\Theta_W \in \mathbb{R}^{K \times C}$ are the weights in the linear layer. The simplified design of $\Theta$ makes estimating importance of each concept function $\phi_k$ for any prediction straightforward.

**Viewability property**   In order to improve visualization for unsupervised CoINs, and thus interpretation of learnt concepts, we propose to add the requirement for a *viewability property*. Given an input image $x$, this property requires to be able to reconstruct *high-quality* images from $\Phi(x)$. Specifically, reconstructions should have high enough quality to "view" input samples through generated outputs and thus ground modifications to $\Phi(x)$ back to $x$. We propose to achieve this by using a pretrained generative model $G$ and learning an additional *concept translator* module $\Omega$ to map $\Phi(x)$ to the latent space of $G$, such that high-quality reconstructed images can be obtained from $\Phi(x)$ through $\Omega$ and $G$. This also retains flexibility to design $\Phi$ for instance in choosing number of concepts $K$ according to the problem, regardless of the choice of pretrained $G$.

**Pretrained generative model $G$ as decoder**   In practice, we want our generative model to (i) have a low dimensional latent space, (ii) have a structured latent space that admits meaningful latent traversals, (iii) be able to generate high-quality images for the underlying data distribution. The choice of using a *pretrained* generator instead of simultaneously training is because it significantly lowers training costs, reduces training complexity and improves reusability. We discuss the significance of these properties in relation to various generative architectures in Appendix B. We also experimentally demonstrate the versatility of our method to different generative architectures in Appendix D.

**Concept translator $\Omega$**   Concept representations in CoINs are typically smaller dimensional than latent spaces of generative models as latent spaces encode lot more information about input than needed for classification. We thus learn the concept representation $\Phi(x)$ separately from the latent space of pretrained $G$ and instead learn a concept translator $\Omega$ to map $\Phi(x)$ to latent space of $G$. The design of $\Omega$ depends on the architecture of underlying generative model. In general, it consists of a fully-connected (FC) layer that predicts for each input image $x$, the latent vector $w_x$ from $\Phi(x)$, that will then be used as input for $G$. The computation of the reconstructed input $\tilde{x}$ is given by:

$$\tilde{x} = G(w_x), \quad \text{where } w_x = \Omega(\Phi(x)). \tag{1}$$

We discuss in more technical details, the architectures of each network in Appendix C.

### 3.3 Training losses

Based on the previous discussion about concept-based networks and our proposed reconstruction pipeline design, we define here our training loss $\mathcal{L}_{train}$ and each of its constitutive terms.

- For the *fidelity to output* property, we define an *output fidelity loss* $\mathcal{L}_{of}$, that grants predictive capabilities to $g$. It's defined as generalized cross entropy (CE) between $g$ and $f$:

$$\mathcal{L}_{of}(x; \Psi, \Theta) = \alpha CE(g(x), f(x)). \tag{2}$$

- The most critical part of our training loss is the *reconstruction loss* $\mathcal{L}_{rec}^{G}$ computed through the pretrained generative model $G$, that gathers all constraints between inputs $x$ and their reconstruction $\tilde{x} = G(\Omega(\Phi(x)))$. It combines $\ell_1$ and $\ell_2$ penalties, enforcing pixel-wise reconstruction for *fidelity to input*, with perceptual similarity LPIPS (Zhang et al., 2018b) and a final *reconstruction classification* term, both linked to *viewability*. The *reconstruction classification* term, defined as $CE(f(\tilde{x}), f(x))$, encourages the generative model to reconstruct $\tilde{x}$ with more classification specific features pertaining to input $x$. Similar losses have been introduced for inversion in generative model and its training for *post-hoc* interpretation (Lang et al., 2021). Our reconstruction loss is thus defined as follows:

$$\mathcal{L}_{rec}^{G}(x; \Psi, \Omega) = ||\tilde{x} - x||_2^2 + ||\tilde{x} - x||_1 + \beta \text{LPIPS}(\tilde{x}, x) + \gamma CE(f(\tilde{x}), f(x)). \tag{3}$$

- We impose the *sparsity* property along with two other regularizations, combined under the term $\mathcal{L}_{reg}$. More specifically, $\Psi$ is regularized to encourage *sparsity* of activations in $\Phi(x)$ through an $\ell_1$ penalty, and *diversity* while reducing *redundancy* in learnt dictionary $\Phi$ with a *kernel orthogonality* loss $\mathcal{L}_{orth}$, applied on weights of final convolution layer of $\Psi$ (Xie et al., 2017; Wang et al., 2020). Then, $\Omega$ is encouraged to predict latent vectors close to average latent vector $\bar{w}$, a common practice in inversion systems of generative models (Tov et al., 2021). The regularization terms are written as follows:

$$\begin{aligned} \mathcal{L}_{reg}(x; \Psi, \Omega) &= \mathcal{L}_{reg-\Psi}(x; \Psi) + \mathcal{L}_{reg-\Omega}(x; \Omega), \\ \mathcal{L}_{reg-\Omega}(x; \Omega) &= ||w_x - \bar{w}||_2^2, \quad \mathcal{L}_{reg-\Psi}(x; \Psi) = \delta ||\Phi(x)||_1 + \mathcal{L}_{orth}(\Psi). \end{aligned} \tag{4}$$

Finally, the training loss and the optimization can be summarized as:

$$\begin{aligned} \mathcal{L}_{train}(x; \Psi, \Theta, \Omega) &= \mathcal{L}_{of}(x; \Psi, \Theta) + \mathcal{L}_{rec}^{G}(x; \Psi, \Omega) + \mathcal{L}_{reg}(x; \Psi, \Omega), \\ \hat{\Psi}, \hat{\Theta}, \hat{\Omega} &= \arg \min_{\Psi, \Theta, \Omega} \frac{1}{N} \sum_{x \in \mathcal{S}} \mathcal{L}_{train}(x; \Psi, \Theta, \Omega). \end{aligned} \tag{5}$$

In the above equations, the loss hyperparameters are denoted by $\alpha, \beta, \gamma, \delta$. During training, $\mathcal{L}_{train}$ is simultaneously optimized w.r.t parameters of $\Psi, \Theta$ and $\Omega$, while keeping $f$ and $G$ fixed.

### 3.4 Interpretation phase

The interpretation generation process can be divided in two parts. (1) **Concept relevance estimation**, that requires estimating the importance of any given concept function $\phi_k$ in prediction for a particular sample $x$ (local interpretation) or a class $c$ in general (global interpretation), and (2) **Concept visualization**, which pertains to visualizing the concept encoded by any given concept function $\phi_k$. We describe each of them in greater detail below:

**(1) Concept relevance:** Since our $g(x)$ adheres to structure of CoINs and among them closest to Parekh et al. (2021), the first step of relevance estimation almost follows as is. The estimation is based on concept activations $\Phi(x)$, and how the pooled version of $\Phi(x)$ is combined by the fully connected layer in $\Theta$ (with weights $\Theta_W$) to obtain the output logits. Note that this step does not rely on using the decoder/generator $G$. Specifically, the local relevance $r_k(x)$ of a concept function $\phi_k$ for a given sample $x$ is computed as the normalized version (between $[-1, 1]$) of its contribution to logit of the predicted class $\hat{c} = g(x)$. The global relevance of concept function $\phi_k$ for a given class $c$, denoted $r_{k,c}$, is computed as the average of local relevance $r_k(x)$ for samples from class $c$. The above description is summarized in equation below wherein $\Theta_W^{k,\hat{c}}$ denotes the weight on concept $k$ for predicted class $\hat{c}$ in weight matrix $\Theta_W$:

$$r_k(x) = \frac{\alpha_k(x)}{\max_l |\alpha_l(x)|}, \quad \alpha_k(x) = \text{pool}(\phi_k(x)) \Theta_W^{k,\hat{c}}, \quad r_{k,c} = \mathbb{E}(r_k(x) | g(x) = c)$$

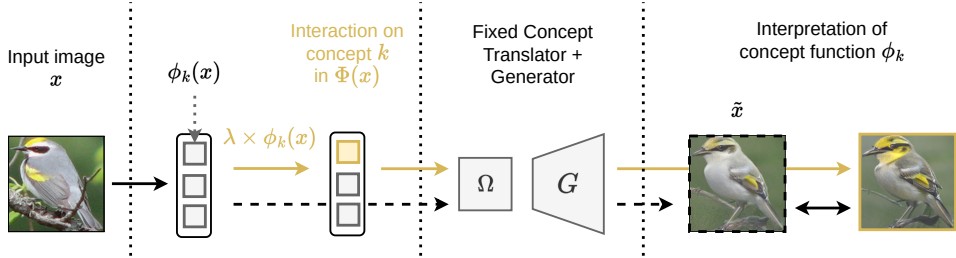

Figure 3: **Visualization for a given image $x$ and concept function $\phi_k$.** By imputing a higher activation for $\phi_k(x)$ in $\Phi(x)$ (by a factor $\lambda = 4$ in the figure), and comparing the obtained visualization to the original reconstruction $\tilde{x}$ (obtained with the untouched $\Phi(x)$), we interpret information encoded by $\phi_k$ about image $x$.

**(2) Concept visualization:** Once the importance of a concept function is estimated, one can extract the most important concepts for a sample $x$ or class $c$ by thresholding $r_k(x)$ or $r_{k,c}$ respectively. For visualizing any concept $\phi_k$, following previous CoINs, one can start by selecting most activating training samples for $\phi_k(x)$ over the whole training set or separately for each class it is highly relevant for. However, the core of our concept visualization process, is to utilize generator $G$ to visualize the impact of $\phi_k$ on any input $x$ it is relevant for. We do so by (1) directly modifying activation of $\phi_k(x)$ by a factor $\lambda \times \phi_k(x), \lambda \geq 0$ while keeping all other activations in $\Phi(x)$ intact, and (2) Visualizing generated output for increasing value of $\lambda$. The case of $\lambda = 1$ corresponds to $\tilde{x}$, the reconstructed version of $x$. This process is summarized in Fig. 3. Extremely high $\lambda$ can push the predicted latent vectors far from the average latent vector in which case the generated output is less reliable. For our experiments, qualitatively we found $\lambda$ upto 3 or 4 reliable with consistent modifications.

## 4 EXPERIMENTS AND RESULTS

**Datasets**  We experiment on image recognition tasks for large-scale images in three different domains with a greater focus on multi-class classification tasks: (1) Binary age classification (young/old) on CelebA-HQ (Karras et al., 2018), (2) fine-grained bird classification for 200 classes on Caltech-UCSD-Birds-200 (CUB-200) (Wah et al., 2011), and (3) fine-grained car model classification of 196 classes on Stanford Cars (Krause et al., 2013).

**Implementation details**  Experiments in main text use a ResNet50 (He et al., 2016) as our architecture for $f$, and StyleGAN2-ADA (Karras et al., 2020) for $G$. Experiments with different architectures for $G$ (ProgressiveGAN (Karras et al., 2018), $\beta$-VAE (Higgins et al., 2017) on FashionMNIST (Xiao et al., 2017)) and $f$ (ResNet101) can be found in Appendix D. All images are processed at resolution $256 \times 256$. We use a dictionary size $K = 64$ on CelebA-HQ and $K = 256$ on CUB-200 and Stanford-Cars. All experiments were conducted on a **single V100-32GB GPU**. Complete details about network architectures, obtaining pretrained checkpoints for $f, G$, VisCoIN training and evaluation metric implementations are provided in Appendix C.

### 4.1 EVALUATION STRATEGY

One major goal of by-design interpretable architectures is to obtain high prediction performance. **Prediction accuracy** of $g$ is thus the first metric we evaluate. We next discuss multiple functionally-grounded metrics (Doshi-Velez & Kim, 2017) that evaluate the learnt concept dictionary $\Phi$ from an interpretability perspective and its use in visualization, including **two novel metrics in the context of evaluating CoINs** ("faithfulness" and "consistency").

**Fidelity of reconstruction**  Since reconstructed output plays a crucial role in our visualization pipeline, we evaluate how well does $G$ reconstruct the input. We compute averaged per-sample mean squared error (MSE), perceptual distance (LPIPS) (Zhang et al., 2018b) and distance of overall distributions (FID) (Heusel et al., 2017) of reconstructed images $\tilde{x}$ and original input images $x$.

**Faithfulness of concept dictionary $\Phi$**  The aspect of *faithfulness* for a generic interpretation method asks the question "are the features identified in the interpretation *truly* relevant for the

prediction process?" (Alvarez-Melis & Jaakkola, 2018a; Parekh et al., 2022). This is generally computed via simulating "feature removal" from the input and observing the change in predictor's output (Hedström et al., 2023). Simulating feature removal from input is relatively straightforward for saliency methods compared to concept-based methods, for example by setting the pixel value to 0. For CoIN systems, this is significantly more tricky, as concept activations $\phi_k(x)$ don't represent the input $x$ exactly. However, through the decoder, we can evaluate if the concepts identified as relevant for an input encode information that is important for prediction. We adopt an approach similar to a previous proposal of faithfulness evaluation for audio interpretation systems (Parekh et al., 2022). Concretely, for a given sample $x$ with activation $\Phi(x)$, predicted class $\hat{c}$ and a threshold $\tau$, we first manipulate $\Phi(x)$ such that $\phi_k(x)$ is set to 0 if $r_k(x) > \tau, \forall k \in \{1, ..., K\}$. That is, we "remove" all concepts with relevance greater than some threshold. This modified version of $\Phi(x)$ is referred to as $\Phi_{rem}(x)$. To compute faithfulness for a given $x$, denoted by FF$_x$, we compute the change in probability of the predicted class from original reconstructed sample $\tilde{x} = G(\Omega(\Phi(x)))$ to new sample $x_{rem} = G(\Omega(\Phi_{rem}(x)))$, that is, FF$_x = g(\tilde{x})_{\hat{c}} - g(x_{rem})_{\hat{c}}$. Ideally, we expect to see a drop in probability (FF$_x > 0$) if the set of relevant concepts "truly" encode information relevant for classification. Following Parekh et al. (2022) we report the median of FF$_x$ over the test data for different thresholds $0 < \tau < 1$.

**Consistency of concept visualization** We expect during visualization of a given concept $\phi_k$ that a user observes similar semantic modifications across different images. Thus, we hypothesize that if modifying any specific concept activation $\phi_k(x)$ leads to consistent changes for different samples $x$, then generated output for two versions of $\Phi(x)$, one with $\phi_k(x)$ set to a large value and one with $\phi_k(x) = 0$, should be separable in the embedding space of $f$ (all other concept activations unchanged). In other words, embeddings for images with high $\phi_k(x)$ and low $\phi_k(x)$ should be well separated. To compute this metric, we first create a dataset of generated images with two different sets of activations. For each of training and test set, this is done by first selecting a set of $N_{cc}$ samples for which $\phi_k$ is highly activating and relevant for. Then we find its maximum activation $\phi_k^{max}$ among these samples, and create two generated outputs for each of $N_{cc}$ samples, one $x_k^+$ such that $\phi_k(x_k^+)$ is set to $\lambda\phi_k^{max}$ with $\lambda \geq 1$, and the other $x_k^-$ such that $\phi_k(x_k^-) = 0$. The two sets of generated images are then gathered into a single dataset $\mathcal{S}_k = \{(x_k^+, 1), (x_k^-, 0)\}$ such that $|\mathcal{S}_k| = 2N_{cc}$, and we learn a binary classifier $\varphi_k : \mathcal{X} \rightarrow \{0, 1\}$, from pooled feature maps of intermediate embedding of $f$. We train for binary classification for sets created from training data $\mathcal{S}_k^{\text{train}}$ and test on sets created from test data $\mathcal{S}_k^{\text{test}}$. Our *concept consistency* metric $CC_k$ for a given concept $k$ is thus obtained as the accuracy of the binary classifier on $\mathcal{S}_k^{\text{test}}$:

$$CC_k(\mathcal{S}_k^{\text{test}}; \varphi_k) = \frac{1}{2N_{cc}} \sum_{(x_k^+, x_k^-) \in \mathcal{S}_k^{\text{test}}} \varphi_k(x_k^+) + (1 - \varphi_k(x_k^-)) \tag{6}$$

This performance is tabulated for each concept $k$ for a fixed $\lambda$, and mean and standard deviation across all concepts is reported.

**Baselines** The primary comparison methods for us are CoINs that learn unsupervised concepts efficiently, even for large-scale images, FLINT (Parekh et al., 2021) and FLAEM (Sarkar et al., 2022). SENN (Alvarez-Melis & Jaakkola, 2018a) suffers from computational issues for large-scale images as it requires to compute jacobian of concept dictionary w.r.t input pixels for its loss computation. Thus, for all the metrics we compare with FLINT and FLAEM as our primary baselines. Additionally, for accuracy evaluation, we track the performance of our pretrained classifier $f$. Note that $f$ (ResNet50) is not an interpretable model and trained entirely for accuracy. Lastly, for faithfulness we compare with a "random" baseline that randomly selects concepts for whom activation is set to 0. Since there is no notion of threshold in selection, in order to make it comparable for a given threshold, we select the same number of concepts randomly as we would for our method. Our proposed system for all evaluations is abbreviated as VisCoIN (Visualizable CoIN).

## 4.2 RESULTS AND DISCUSSION

**Quantitative results** Table 1a reports the test accuracy of all the evaluated systems. Our proposed system, VisCoIN performs competitively with the pretrained $f$ considered uninterpretable and purely trained for performance. It also performs better than the other recent CoIN systems for more complex classification tasks (CUB-200 and Stanford Cars) with large number of classes and diverse images. Metrics quantifying the fidelity of reconstruction on test data are in Table 2a. The other baselines only optimize for pixel-wise reconstruction and FLINT achieves a lower MSE than

Table 1: **(a)** Accuracy (in %) of interpreter $g$ of CoIN systems, and of the baseline pretrained classifier $f$. **(b)** Mean and standard deviation for consistency $CC_k$ over all concept functions $\phi_k$ (binary accuracy in %). Higher is better, the best performance is reported in **bold**, second best in underline.

(a) Accuracy of interpreter $g$

| Dataset | Original-$f$ | FLINT | FLAEM | VisCoIN (Ours) |
|---|---|---|---|---|
| CelebA-HQ | 87.71 | 87.25 | **88.18** | 87.71 |
| CUB-200 | **80.56** | 77.2 | 51.76 | 79.44 |
| Stanford Cars | **82.28** | 75.95 | 50.02 | 79.89 |

(b) Consistency of changes

| Dataset | FLINT | FLAEM | VisCoIN (Ours) |
|---|---|---|---|
| CelebA-HQ | 82.6 ± 22.7 | 57 ± 17.3 | **85.5 ± 13.9** |
| CUB-200 | 72.6 ± 18 | 55.6 ± 13.6 | **85 ± 8.4** |
| Stanford Cars | 70 ± 16.3 | 54.9 ± 13.3 | **82.7 ± 8.3** |

Table 2: **(a)** Reconstruction quality (MSE, LPIPS and FID) of CoIN systems. Lower is better. **(b)** Faithfulness (median $FF_x$) of CoIN systems and random baseline, for different threshold. Higher is better. Best performance is in **bold**, second best in underline.

(a) Reconstruction quality

| Dataset | Metric | FLINT | FLAEM | VisCoIN (Ours) |
|---|---|---|---|---|
| CelebA-HQ | MSE | **0.051** | 0.119 | 0.094 |
| | LPIPS | 0.533 | 0.688 | **0.405** |
| | FID | 30.45 | 39.73 | **8.55** |
| CUB-200 | MSE | **0.113** | 0.217 | 0.161 |
| | LPIPS | 0.712 | 0.75 | **0.545** |
| | FID | 53.16 | 51.15 | **15.85** |
| Stanford Cars | MSE | **0.121** | 0.278 | 0.179 |
| | LPIPS | 0.697 | 0.734 | **0.488** |
| | FID | 64.16 | 69.44 | **6.77** |

(b) Faithfulness

| Dataset | Thresh. $\tau$ | Random | FLINT | FLAEM | VisCoIN (Ours) |
|---|---|---|---|---|---|
| CelebA-HQ | 0.1 | 0.03 | 0.254 | 0.091 | **0.267** |
| | 0.2 | 0.018 | **0.201** | 0.151 | 0.171 |
| | 0.4 | 0.005 | 0.07 | **0.107** | 0.074 |
| CUB-200 | 0.1 | 0.034 | 0.004 | $< 10^{-3}$ | **0.251** |
| | 0.2 | 0.007 | 0.002 | $< 10^{-3}$ | **0.146** |
| | 0.4 | 0.001 | $< 10^{-3}$ | $< 10^{-3}$ | **0.044** |
| Stanford Cars | 0.1 | 0.035 | 0.001 | $< 10^{-3}$ | **0.161** |
| | 0.2 | 0.016 | $< 10^{-3}$ | $< 10^{-3}$ | **0.118** |
| | 0.4 | 0.002 | $< 10^{-3}$ | $< 10^{-3}$ | **0.034** |

VisCoIN. However, crucially, reconstruction from our method approximates the input data considerably better, in terms of perceptual similarity (LPIPS) and overall distribution (FID), which highly contributes to better viewability. Table 2b tabulates the median faithfulness $FF_x$ for the evaluated systems on 1000 random test samples for different thresholds. The performance of *Random* baseline being close to 0 even for small thresholds indicates that a random selection of concepts often contains little information relevant for classification of the predicted class. In contrast, concepts identified relevant as part of $g$ in VisCoIN tend to encode information about input that noticeably affects classification. In regard to other CoINs, while the faithfulness results are competitive on CelebA-HQ, for more complex datasets, concept dictionary in VisCoIN is significantly more faithful than FLINT or FLAEM, which do not demonstrate more faithfulness than the *Random* baseline. Finally, the mean and standard deviation for visualization consistency of all concepts is reported in Table 1b with $\lambda = 2$. Concepts learnt with VisCoIN demonstrate a higher mean consistency of visualization compared to baselines. The deviation across concepts is also lower for our method. We also evaluate concept consistency with higher values of $\lambda$ and observe increased separation with better classification performance (Appendix E).

**Qualitative results** Fig. 4 shows visualization for different class-concept pairs across the three datasets that are determined to have high global relevance $r_{k,c}$ through predictive structure of $g(x)$, as described in Section 3.4. For each class-concept pair, we show two maximum activating training samples for the concept from the corresponding class, the reconstructed input from $\Phi(x)$ ($\lambda = 1$) and the generated output with modified concept activation $\phi_k(x)$ by a factor $\lambda = 4$. In all the illustrations, increasing the activation of the concept, *i.e.* moving from $\lambda = 1$ to 4, strongly emphasizes some specific concept in the generated output that can be clearly grounded to the input, and the name of the concept is then manually inferred. For instance, increasing the activation of concept for "Red-eye" in Fig. 4a increases the size of red eye of the bird, a key feature of samples from class 25 ("Bronzed-cowbird"). We can also qualitatively verify that the reconstruction has a high enough quality that allows us to "view" the input sample through the generated output and ground the modifications in generated output to the input. However, we also observed that learnt concept functions can be prone to modifying more than one high-level feature in the image. For, *e.g.*, in Fig. 4d, increasing the concept activation increases both "eye-squint" and "beard" in the generated output. A longer discussion about limitations is available in Appendix H.

**Ablation study** We ablate multiple aspects of our system, with detailed results in Appendix F. Notably, we observed a tradeoff induced by strength of reconstruction-classification loss (weight $\gamma$). A high $\gamma$ positively impacts faithfulness, but negatively impacts perceptual similarity of reconstruction.

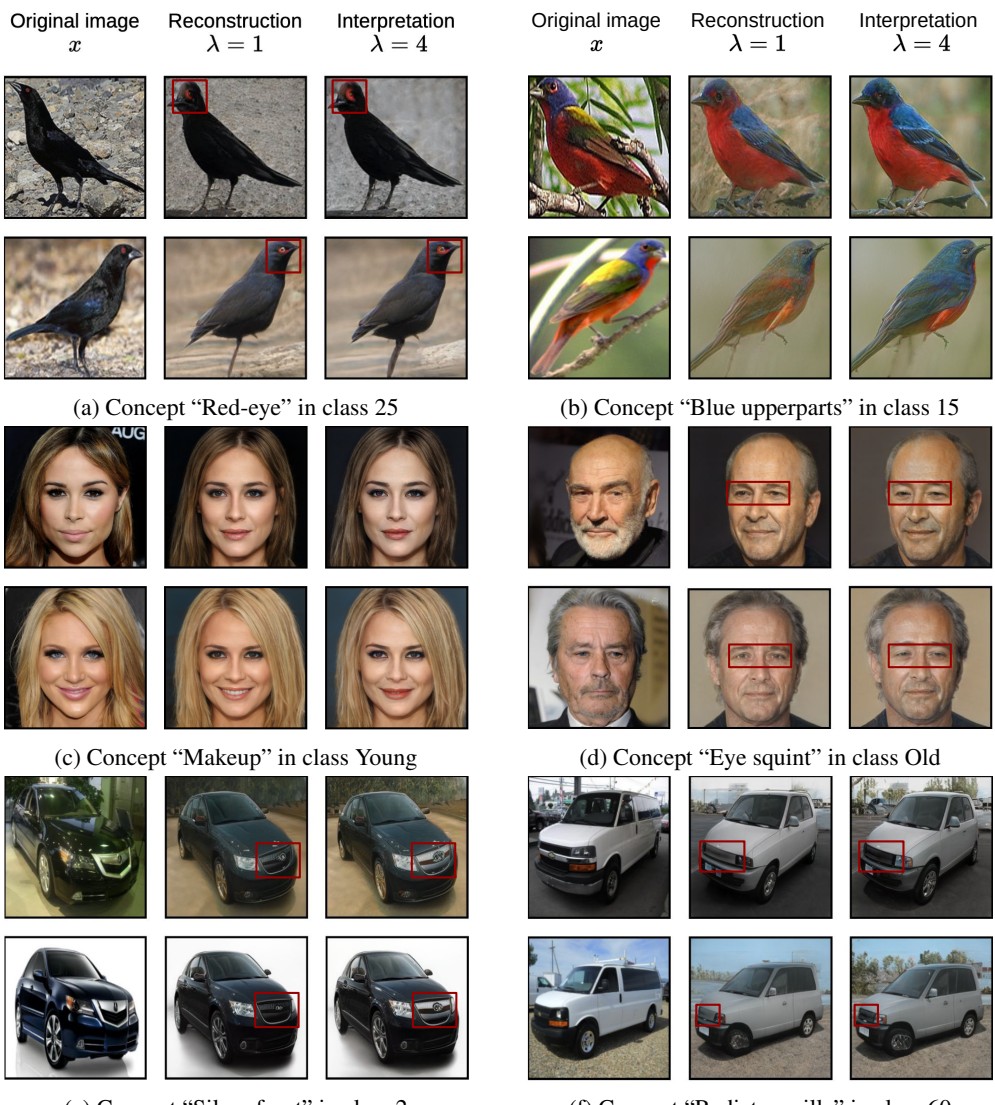

Figure 4: Qualitative examples obtained for different concepts, classes on **(a)-(b)** CUB-200, **(c)-(d)** CelebA-HQ, **(e)-(f)** Stanford-Cars datasets. On each subfigure, first column corresponds to maximum activated samples $x$ for class-concept pairs with high relevance ($r_{k,c} > 0.5$), second column to reconstructed image obtained with original $\Phi(x)$, and third column to the image obtained by imputing $4 \times \phi_k(x)$ in $\Phi(x)$. Red boxes manually added to indicate key regions of modifications in generated images.

## 5    CONCLUSION

We introduced a novel architecture for *Visualizable CoIN (VisCoIN)*, that addresses major limitations to visualize unsupervised concept dictionaries learnt in CoIN systems for large-scale images. Our architecture integrates the visualization process in the pipeline of the model training, by leveraging a pretrained generative model using a *concept translator* module. This module maps concept representation to the latent space of the fixed generative model. During training, we additionally enforce a viewability property that promotes reconstruction of high-quality images through the generative model. Finally, we defined new evaluation metrics for this novel interpretation pipeline, to better align evaluation of concept dictionaries and interpretations provided to a user. Future works include adapting the design of this system for supervised CoINs, multimodal generative models, or extending its application to different data modalities.

## Reproducibility Statement

Throughout the paper, we made sure that all our experiments were fully reproducible, describing in details all datasets and architectures considered in Section 4. We then explain the design of $\Psi$ and $\Omega$, training settings, hyperparameters and computation of evaluation metrics in Appendix C.

## Acknowledgements

This work was supported by the LIMPID (ANR-20-CE23-0028) and FAR-SEE (ANR-24-CE23-0921) projects of the French National Research Agency (ANR). The authors also thank Thibaut de Saivre and Hugo Aoyagi for restructuring the codebase.

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

The Appendix is organized as follow:

- We discuss relation with CBMs (supervised CoINs) using language models to extract concepts annotations, and the general usefulness of concept visualization in Appendix A.

- We discuss the desidered properties for the generative model $G$ in Appendix B.

- We describe in details the architectures of the different network, the training procedures of the system and the evaluations in Appendix C.

- We present additional experiments with different architectures for $f$ and $G$, to showcase the flexibility of the system in Appendix D.

- We present additional evaluation and comparisons with other unsupervised CoINs in Appendix E.

- We present ablation studies and discussions about design selection of the different components of the system in Appendix F.

- We show additional visualizations for qualitative analysis in Appendix G.

- We discuss overall limitations of the system in Appendix H and potential negative impacts in Appendix I.

## A  LANGUAGE-BASED CONCEPTS

### A.1  SUPERVISED COINS AND LANGUAGE MODEL BASED CBMS

Recent variants of concept bottleneck models (CBMs) are trained with concept sets generated from language models (Yuksekgonul et al., 2022; Oikarinen et al., 2023; Yang et al., 2023b; Panousis et al., 2024). These models are also CoINs that do not require human annotations for training. They obtain sample-wise concept annotations using concept set constructed from language models. Specifically, for each class, they prompt a language model to generate a set of text descriptions describing the class. This set of descriptions is then used as the concept bank. For each sample, the concept annotation during training is obtained by CLIP similarity between the image and text descriptions of its ground-truth class. Note that all of these steps are executed prior to training the concept bottleneck model.

Interestingly, while these recent CBMs provide text descriptions for concepts thanks to automatic extraction from LLMs, they are even worse offenders of the visualization limitations highlighted in Section 1. This is because they currently do not have any reliable visualization pipeline to confirm if the visual detection of a given concept by the underlying CBM corresponds to its text description or not. Moreover, in many cases the concepts as part of concept annotations are not even visually grounded (e.g. "Loyal/honest" for class "Dog") in which case it is impossible to visualize the concept in the first place. However, they are methodologically very close to original CBMs or supervised CoINs as they still use sample-wise annotations to train $\Phi(x)$.

Besides the use of concept labels, CBMs do not have a decoder model. There is no way to conduct any of the main evaluation (reconstruction, faithfulness, consistency) without a decoder, since we need to approximate the input from concept activations to do any of these. All unsupervised CoINs have a decoder and none of the CBMs do, as a result of their training methodological differences. Thus, they are currently out of scope for VisCoIN which is focused more on unsupervised CoINs.

### A.2  ON THE USEFULNESS OF VISUALIZATION

We present below arguments why visualizing concepts is still important:

- When considering expert or domain specific datasets like Stanford Cars (Krause et al., 2013) (for car models classification) or the MVTec Anomaly Detection (Bergmann et al., 2021) (for anomaly detection of object in production lines) for instance, visualizing concepts directly on the objects is simpler and faster to understand for human operators, rather than reading a text description.

- For certain computer vision applications (eg. self-driving cars, medical imaging tasks), visualization provides spatially localized interpretations, which is more difficult and cumbersome with text. For instance, if a concept relating to "red light" is activated for an image, to get a thorough understanding of the model's decision, it is crucial to identify which regions and what content in the image activates the concept. Ideally, this would be best proved by a human evaluation comparing language descriptions and visualizations for real-world applications, to evaluate if/how much advantage visualizations add. However, for fair comparison language description and visualization should be of the same concept dictionary for the same model. To design such a study and system remains a challenging problem, which we will explore as a future direction.

- The LLMs/VLMs which the recent CBMs are based on (particularly CLIP (Radford et al., 2021)) are limited when detecting concepts and image details at a finer spatial scale (Gou et al., 2024).

- As discussed above, the current methods are prone to generating concept descriptions not grounded in any visual information, which also harms their interpretability.

- In the case of LLM/VLM based CBMs, there are also concerns about faithfulness of concept detection to the text description. This is a similar issue to concept leakage (Havasi et al., 2022). We believe that the ideas presented in our work, such as viewability, can help in identifying such issues in LLM/VLM based CBMs.

## B DESIDERATA FOR GENERATIVE MODEL

We discuss below the desired properties that can influence the choice of an appropriate generative model $G$, as previously mentioned in Section 3.2.

(i) The generative model should have the ability to model the input with a low dimensional latent space (compared to input dimensions), since a concept representation is typically much lower dimensional than input.

(ii) It possesses a structured latent space that admits convenient mechanisms for meaningful latent traversal. This directly helps in designing a $\Omega$ such that modifying a concept activation can enable latent traversal for visualization.

(iii) The generative model $G$ is able to unconditionally generate high quality samples, close to the underlying dataset/data distribution. This is essential not only for better reconstruction of images, but also crucial to ground any visual modifications in the generated images back to the original input.

The desiderata discussed above enables our approach to be compatible with a variety of generative models. Notably, most GANs and VAEs satisfy the first two requirements. Provided they are capable to approximate well the underlying data distribution, our approach fits well with them as choice of $G$. The experiments in main text focus on StyleGAN2-ADA (Karras et al., 2020) as our $G$ architecture given its high-quality generation for various large-scale image domains. We also illustrate applicability of our method of other $G$ architectures such as ProgressiveGAN (Karras et al., 2018) and $\beta$-VAE (Higgins et al., 2017) in Appendix D.

For diffusion model (Sohl-Dickstein et al., 2015) architectures, despite the recent positive steps towards understanding their latent space (Kwon et al., 2023; Park et al., 2023), it is currently difficult to design an $\Omega$ that allows a straightforward and meaningful latent traversal. Nevertheless, it is worth mentioning that latent diffusion models (Rombach et al., 2022) do satisfy (i) and (iii) in many cases. With further research in understanding their latent spaces, we believe it could be possible to satisfy (ii) and extend VisCoIN to them.

## C  FURTHER SYSTEM DETAILS

### C.1  NETWORK ARCHITECTURES

The networks $f$ and $G$ are pretrained and fixed during training of VisCoIN. As part of our training, we train three subnetworks $\Psi, \Omega, \Theta$. We already described $\Theta$ in the main text, as consisting of a pooling (maxpool), linear and softmax layers in the respective order.

**General Architecture of $\Psi$**  Our architecture of $\Psi$ mostly follows proposed architecture of $\Psi$ for FLINT (Parekh et al., 2021) which accesses output of two layers for ResNet18 close to the output layer (output of block 3 and penultimate layer of block 4). The ResNet50 also follows a similar structure with 4 blocks. Each block however contains 3, 4, 3 and 3 sub-blocks termed "bottleneck" respectively. In terms of the set of layers accessed by $\Psi$ for VisCoIN, in addition to the corresponding two layers in ResNet50 (output of block 3 of shape $1024 \times 16 \times 16$, output of penultimate bottleneck layer in block 4 of shape $2048 \times 8 \times 8$), we also access a third layer for improved reconstruction (output of block 2 of shape $512 \times 32 \times 32$). Each layer output is passed through a convolutional layer and brought to a common shape of $512 \times 8 \times 8$, the lowest resolution and feature maps. We then concatenate all the feature maps to output $\Phi(x)$. We apply two convolutional and a pooling layer yielding an output shape of $K \times 3 \times 3$, where $K$ is the number of concepts, and each $\phi_k(x)$ is a convolutional map of size $3 \times 3$. Thus, the total number of elements in each $\phi_k(x)$ is $b = 9$.

**General Architecture of $\Omega$**  The typical theme we use to implement $\Omega$ is as a single linear layer that takes as input $\Phi(x)$ and outputs a vector in the latent space of the generative model. This directly associates each $\phi_k$ with a vector in the latent space of $G$, specified by the $k$-th column vector of weight matrix in $\Omega$. While this exactly corresponds to our implementation for ProgressiveGAN, $\beta-$VAE, our network designs for StyleGAN2 have slight modifications to $\Psi, \Omega$ even though we still follow the same themes. We discuss next the precise architectures with StyleGAN next.

### C.1.1  RECONSTRUCTION WITH STYLEGAN

The reconstruction architecture with StyleGAN is similar to encoder based GAN-inversion architectures used for StyleGAN. Following previous works in this regard (Abdal et al., 2019; Yao et al., 2022), we use the extended latent space $\mathcal{W}^+$ for inversion, which corresponds to different latent vectors for different resolutions. We learn the concept translator $\Omega$ to map the concept activations to $\mathcal{W}^+$ and bias its computation with average latent vector $\bar{w}$ of $G$.

Since $\Phi(x)$ is a much lower dimensional representation compared to elements in $\mathcal{W}^+$ and constrained by losses unrelated to reconstruction, we found it challenging to achieve reconstruction quality close to GAN inversion methods. This issue in principle cannot be completely eliminated without compromising the interpretable predictive structure. However, we alleviate it by learning an unconstrained "supporting" representation as a secondary output from $\Psi$, termed $\Phi'(x)$. The only goal for $\Phi'(x)$ is to assist $\Omega$ in embedding the input in $\mathcal{W}^+$. To predict $\Phi'(x)$ the concatenated feature maps in $\Psi$ are sent in a second parallel branch, that applies two fully connected layers to output same number of elements as in $\Phi(x)$. The computation for reconstruction $\tilde{x}$ is then given as:

$$\tilde{x} = G(w_x^+), \text{where } w_x^+ = \Omega(\Phi(x), \Phi'(x)) \in \mathcal{W}^+ \tag{7}$$

In this case, the concept translator $\Omega$ consists of a set of single fully-connected (FC) layers, one for predicting each latent vector. Each FC layer either takes $\Phi(x)$ or $\Phi'(x)$ as input depending upon the latent vector it predicts. To determine, which latent vectors should be controlled by $\Phi'$, we rely on findings from the work in Katzir et al. (2022) which roughly divides the different style vectors in $\mathcal{W}^+$ as controlling the coarse, mid and fine level features of the generated image with increasing resolution. For 14 latent vectors in case of $256 \times 256$ output resolution, it corresponds to 4, 4, 6 vectors respectively. We expect the relevant features for classification to be controlled mostly by mid-level and fine-level style vectors, except possibly for the highest resolution where very fine-scaled details are controlled. Thus, we predict the first three and last two style vectors using $\Phi'(x)$. The rest of the style vectors (9 out of 14 for resolution 256) are predicted from $\Phi(x)$. The choice of using $\Phi'$ is analyzed with an ablation study in Table 15.

## C.2 TRAINING DETAILS

The steps to train our system on a given dataset can be divided into three modular parts: (1) Obtaining a pretrained classifier $f$ with "strong" performance that can provide high-quality source representations to learn from, (2) Obtaining a pretrained generator $G$ that can approximate well the distribution of the given dataset, and (3) Training of $g$ with VisCoIN using the pretrained $f$ and $G$. When a pretrained $f$ or $G$ is not easily available, we train them on their respective tasks on the given dataset. Among our 3 datasets, CelebA-HQ (Karras et al., 2018), CUB-200 (Wah et al., 2011) and Stanford Cars (Krause et al., 2013), we easily found a pretrained $G$ for CelebA-HQ. All other combinations of $f$ and $G$ were pretrained. We describe the training details of $f$, $G$ and VisCoIN below.

### C.2.1 PRETRAINING $f$

We pretrain $f$ for classification on each of our datasets before using it for training VisCoIN. We use Adam optimizer (Kingma & Ba, 2014) with fixed learning rate 0.0001 on CUB-200 and 0.001 on CelebA-HQ to train $f$. On Stanford-Cars, we use SGD optimizer with a starting learning rate of 0.1, decayed by a factor 0.1 after 30 and 60 epochs. The training is initialized with pretrained weights from ImageNet in each case, and fine-tuned for 10, 30 and 90 epochs on CelebA-HQ, CUB-200 and Stanford-Cars respectively. In all cases, during pretraining, the images are resized to size $256 \times 256$. The accuracy of $f$ is already reported in the main paper. All of these experiments have been conducted on a single A100 GPU, with a batch size of 64 for CelebA-HQ and 128 for Stanford-Cars dataset and on V100 GPU with a batch size of 32 for CUB-200.

### C.2.2 PRETRAINING $G$

We use a pretrained StyleGAN2-ADA (Karras et al., 2020) for experiments in the main text. On CelebA-HQ, we used a pretrained checkpoint available from NVIDIA. For CUB-200 and Stanford-Cars, we pretrain $G$ ourselves. Note that since we want $G$ to generate images entirely from information provided by $\Phi(x)$, we do not use any class labels when training $G$. We use the official StyleGAN2-ADA Pytorch repository to train our models. A challenge that can arise is from limitations to training resources since these models might require to be trained with tens of millions of real/dataset images ("shown" to the discriminator) in order to reach high quality generation. This could potentially require training with multiple GPUs for multiple days. We address this issue to a reasonable extent by fine-tuning pretrained checkpoints. We utilize the insights from (Grigoryev et al., 2022) and fine-tune a checkpoint from ImageNet for CUB-200, and LSUN Cars (Yu et al., 2015) for Stanford Cars. The choices of these specific models was specifically based on the idea that these datasets were the closest domains we had access of pretrained checkpoints to. We also present experiments with pretrained ProgressiveGAN (Karras et al., 2018) and $\beta$-VAE (Higgins et al., 2017) in Appendix D. For ProgressiveGAN, we use a pretrained checkpoint on CelebA-HQ provided by NVIDIA. The $\beta$-VAE is trained by us with $\beta = 2$.

We train the $G$ on a single Tesla V100-32GB GPU with mostly default parameters from the official repository. We only differ in (1) Learning a mapping function (that learns to predict latent vectors from gaussian noise vector) with 2 FC layers and (2) For Stanford cars, we observed a collapse in generation of viewpoints with default training after 600k images shown, thus we reduced the strength of horizontal flip augmentation to 0.1 instead of default 1.

We use a batch size of 16 for training. The GANs are trained only on the training data. The final pretrained model for CUB-200 is obtained after training the discriminator with 2 million real images (21 hours). The final model for Stanford-Cars was obtained after training with 1.8 million dataset images (18.5 hours). The pretrained models achieve an FID of around 9.4 and 8.3 on CUB-200 and Stanford Cars, respectively.

### C.2.3 TRAINING VISCOIN

We train for 50K iterations on CelebA-HQ and 100K iterations on CUB-200 and Stanford-Cars. We use Adam optimizer with learning rate 0.0001 for all subnetworks and on all datasets. During training, each batch consists of 8 samples from the training data and 8 synthetic samples randomly

---

https://github.com/NVlabs/stylegan2-ada-pytorch

generated using $G$. This practice of utilizing the synthetic samples from $G$ is fairly common for encoder-based GAN inversion systems (Yao et al., 2022; Xia et al., 2022), and an additional advantage for our system to use a pretrained $G$. Note that the use of fidelity loss with a pretrained $f$ instead of a classification loss on $g(x)$ fits neatly with this, as one cannot obtain any ground-truth annotations for the synthetic samples. The training data samples use a random cropping and random horizontal flip augmentation in all cases. All images are normalized to the range $[-1, 1]$ and have resolution $256 \times 256$ for processing. This is the default range and resolution we use for pretraining for $f$ and $G$ too. We have already described the architectures of all our components, pretrained or trained as part of training VisCoIN. We tabulate below in Table 3 the hyperparameter values for all our datasets. To limit the amount of hyperparameters to tune, we used a fixed $\alpha = 0.5$ (weight for output fidelity loss) and $\beta = 3$ (weight for LPIPS loss) for all datasets. The rationale behind choice of all hyperparameters is discussed in Appendix F, wherein we also present the ablation studies w.r.t to multiple components.

Table 3: Hyperparameters values for VisCoIN

| Parameter | CelebA-HQ | CUB-200 | Stanford Cars |
|---|---|---|---|
| $K$ – Size of concept dictionary $\Phi$ | 64 | 256 | 256 |
| $\alpha$ – Weight for output fidelity | 0.5 | 0.5 | 0.5 |
| $\beta$ – Weight for LPIPS | 3.0 | 3.0 | 3.0 |
| $\gamma$ – Weight for reconstruction-classification | 0.2 | 0.1 | 0.05 |
| $\delta$ – Weight for sparsity | 2 | 0.2 | 0.2 |

## C.3 EVALUATION DETAILS

### C.3.1 METRIC COMPUTATION

The median faithfulness is computed over 1000 random samples from the test data. For consistency, we use $N_{cc} = 100, \lambda = 2$, *i.e.*, given any concept $\phi_k$, we extract its 100 most activating samples over samples of classes its most relevant for. The constant high activation is twice ($\lambda = 2$) the maximum activation of $\phi_k(x)$ over the pool of 100 samples. Thus the binary training dataset created via samples from training data consists of 200 samples, 100 "positive" samples with high activation of $\phi_k(x)$ and 100 "negative" samples with zero activation $\phi_k(x)$. The binary testing dataset also contains the same number of samples of each type but is created via samples from test data. The feature maps we extract are the output of the second block of the pretrained $f$ (ResNet50), the 22nd convolutional layer. The shape for each feature map is $512 \times 32 \times 32$. We pool them across the spatial axis to obtain an embedding of size 512 for any input sample. The linear classifier we train is a linear SVM. We select its inverse regularization strength $C$ from the set $\{0.01, 0.1, 1.0, 5.0\}$ (lower value is stronger regularization). The parameter is selected using 5-fold cross validation on the created training data.

### C.3.2 BASELINE IMPLEMENTATIONS

We utilize the official codebase available for FLINT and FLAEM for our baseline implementation. For fairness, we use the same number of concepts for both of them. Since our architecture is closer to FLINT, we update and adapt it to implement in similar settings as ours. We use the same $f$ architecture for both the systems and keep it pretrained and fixed. The $\Psi$ architecture is also similar in that it accesses the same set of hidden layers and has the same structure and depth. For other hyperparameters we use their default settings applied earlier for CUB-200. Implementing FLAEM with same network architecture is more complicated as it deviates considerably from the proposed architecture, thus we mostly use their default settings. In their code, they use a base classifier architecture similar to ResNet101 and use the output of final conv layer as the concept representation. In both cases, we do not modify the decoders. FLAEM uses a simpler decoder that learns 3 deconvolution layers, while FLINT learns a deeper decoder consisting of transposed convolution layers.

---

https://github.com/jayneelparekh/FLINT
https://github.com/anirbansarkar-cs/Ante-hoc_Explainability_Concepts

Table 4: Accuracy of interpreter (in %), reconstruction quality (MSE, LPIPS and FID), faithfulness (median $FF_x$ for threshold $\tau = 0.2$) and consistency $CC_k$ (mean and standard deviation of binary accuracy in %), on CelebA-HQ dataset, when using different generative models as decoder.

| Method | Acc. (↑) | MSE (↓) | LPIPS (↓) | FID (↓) | $FF_x$ (↑) | $CC_k$ (↑) |
|---|---|---|---|---|---|---|
| VisCoIN - ProgressiveGAN | 87.82 | 0.095 | 0.453 | 6.98 | 0.37 | $93.8 \pm 6.8$ |
| VisCoIN - StyleGAN2-ADA | 87.71 | 0.094 | 0.405 | 8.55 | 0.171 | $85.5 \pm 13.9$ |

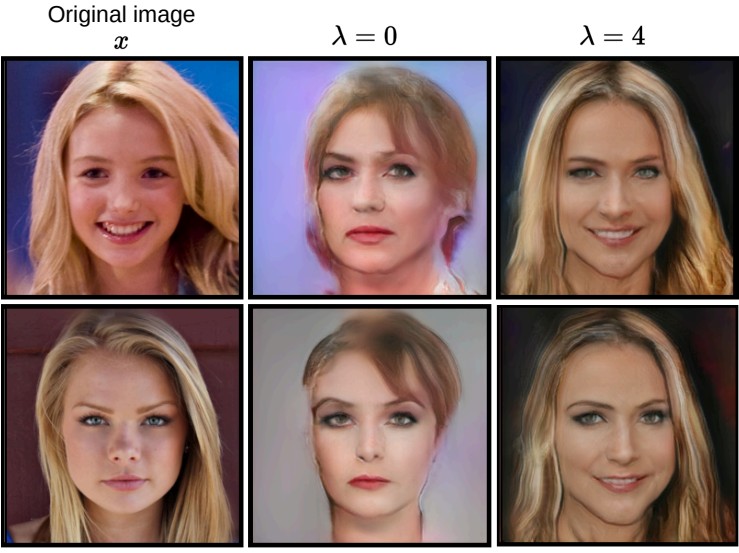

Concept "Smile + long hair" in class Young

Figure 5: Qualitative results of VisCoIN with ProgressiveGAN (Karras et al., 2018) on CelebA-HQ dataset.

## D    EXPERIMENTS WITH OTHER ARCHITECTURES FOR $f$ AND $G$

In this section, we demonstrate that our model can generalize to a variety of network architectures for $f$ and $G$. In particular, we show applicability of VisCoIN with LeNet, ResNet101 and ViT-B/16 architectures for $f$, and $\beta$-VAE and ProgressiveGANs architectures for $G$.

### D.1    EXPERIMENT WITH PROGRESSIVEGAN

We present in Table 4, results of experiments using a pretrained ProgressiveGAN (Karras et al., 2018) as our generative model $G$, on CelebA-HQ dataset, all other hyperparameters being identical as experiments presented in the main text on this dataset. As can be seen in the table, we obtain similar accuracy and MSE, but FID, faithfulness and consistency are better using ProgressiveGAN, while LPIPS is better using StyleGAN2-ADA. We show an example concept visualization in Fig. 5. The visualization clearly reveal two features ('Smile', 'Long Hair') both controlled by a single concept

Table 5: Accuracy of interpreter (in %), reconstruction quality (MSE), faithfulness (median $FF_x$ for different threshold) and consistency $CC_k$ (mean and standard deviation of binary accuracy in %), on FashionMNIST dataset.

| Method | Acc. (↑) | MSE (↓) | $FF_x$ (↑) 0.4 | $FF_x$ (↑) 0.2 | $FF_x$ (↑) 0.1 | $CC_k$ (↑) |
|---|---|---|---|---|---|---|
| VisCoIN - $\beta$-VAE | 88.06 | 0.029 | 0.576 | 0.693 | 0.762 | $94.3 \pm 7.4$ |
| FLINT | 86.41 | 0.031 | 0.564 | 0.688 | 0.728 | $96.2 \pm 4.8$ |

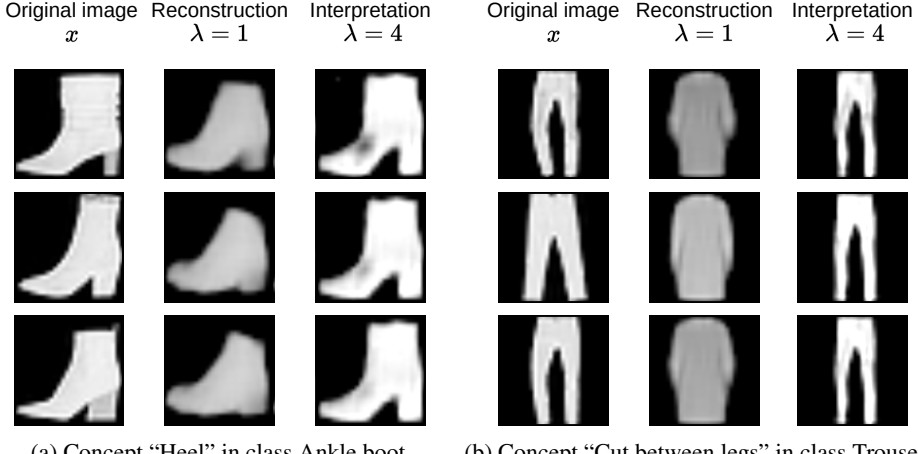

| Original image $x$ | Reconstruction $\lambda = 1$ | Interpretation $\lambda = 4$ | Original image $x$ | Reconstruction $\lambda = 1$ | Interpretation $\lambda = 4$ |

(a) Concept "Heel" in class Ankle boot          (b) Concept "Cut between legs" in class Trousers

Figure 6: Qualitative results for VisCoIN - $\beta$-VAE on FashionMNIST.

Table 6: Accuracy of interpreter (in %), reconstruction quality (MSE, LPIPS and FID), faithfulness (median $FF_x$ for threshold $\tau = 0.2$) and consistency $CC_k$ (mean and standard deviation of binary accuracy in %), on CUB dataset, when using different architectures for $f$.

| Method | Acc. ($\uparrow$) | MSE ($\downarrow$) | LPIPS ($\downarrow$) | FID ($\downarrow$) | $FF_x$ ($\uparrow$) | $CC_k$ ($\uparrow$) |
|---|---|---|---|---|---|---|
| VisCoIN - RN50 | 79.44 | 0.16 | 0.545 | 15.85 | 0.146 | $85.0 \pm 8.4$ |
| VisCoIN - RN101 | 79.44 | 0.16 | **0.542** | **11.24** | **0.202** | **$94.0 \pm 6.2$** |

function. The high consistency and faithfulness is also possibly the result of strong modifications in generated images when traversing latent space in ProgressiveGAN. However, most importantly, reconstruction quality worsens compared to results on StyleGAN and affects visualization. This is a major factor in our preference of the StyleGAN version of VisCoIN over ProgressiveGAN version of VisCoIN. This is also a good example to illustrate the important role viewability plays in visualization.

## D.2   EXPERIMENT WITH BETA-VAE

We also experiment with applying VisCoIN using $\beta$-VAE (Higgins et al., 2017) as generative model, on FashionMNIST dataset (Xiao et al., 2017). We use $K = 25$ as the number of concepts. The VAE is trained with $\beta = 2$ and latent embedding size of 48. Given that the dataset consists of grayscale images of smaller resolution, we do not use LPIPS loss during training. We use a $f$ architecture similar to LeNet (LeCun, 2015).

Table 5 reports the result of our VisCoIN system against FLINT as a baseline. For reconstruction, we only report the MSE since both LPIPS and FID rely on fixed networks pretrained on large scale color images and are thus not suitable for this dataset. We can see that our VisCoIN achieves better accuracy, reconstruction quality and faithfulness than FLINT. For small and grayscale images, even though we observe an advantage in using a generative model over a standard decoder, the effects are less pronounced compared to more complex images and classification tasks (as in main text). The main reason for this is that in the current scenario, the standard decoders are also quite capable at generating viewable reconstructions. This gap with generative models grows larger as the underlying data distributions grows more complex. Nonetheless, crucially, these experiments show that our method is effective even with a completely different generative model. Qualitative results of two concepts and their visualizations can be found in Fig. 6.

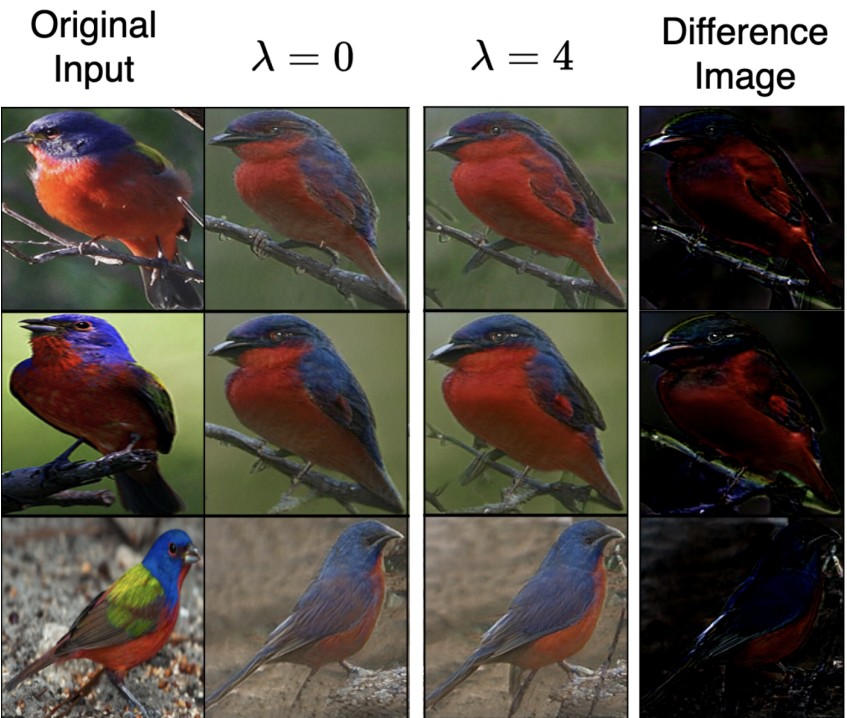

Figure 7: Example concept visualization ($\phi_{36}$, class 15) for ViT-B/16 on CUB-200 ("Red front/belly"). We show the original input, generated/reconstructed images with intervened activations ($\lambda = 0, 4$) and the difference between generated images to localize the modifications.

### D.3    EXPERIMENT WITH RESNET101

We present in Table 6 additional experiments on CUB dataset, when using a ResNet101 network as our classification network $f$, all other hyperparameters being identical. We can see that VisCoIN achieves similar accuracy but better perceptual reconstruction quality (LPIPS and FID), faithfulness and consistency. While relatively nominal, the consistent improvement in all interpretability evaluation metrics compared to ResNet-50 is likely the result of better quality of hidden layers supplied by a larger backbone.

### D.4    EXPERIMENT WITH VIT-B/16

We report in Table 7 an additional experiment using a ViT-B/16 for $f$ on CUB dataset, while keeping other architectures almost identical to experiments in main text. We started from a ViT-B/16, pretrained on ImageNet-21K (Wu et al., 2020; Deng et al., 2009) and only finetuned the classification head on CUB, to use as $f$. We take the patch embeddings of final layer as input to $\Psi$. We keep identical $\Omega$, $\Theta$ hyperparameters, and slightly modify $\Psi$ for reduced number of feature maps. As can be seen from the numerical results below, we achieve better accuracy thanks to a better pretrained $f$, but reconstruction (LPIPS) and faithfulness are worse. The results could be improved by better designing $\Psi$ and accessing more internals embeddings. However, they certainly show that VisCoIN can generalize to other backbone architectures. An example concept visualization to support this is shown in Fig. 7.

Table 7: Accuracy of interpreter (in %), reconstruction quality (LPIPS), faithfulness (median $FF_x$ for threshold $\tau = 0.2$), on CUB-200 dataset, when using ViT-B/16 architecture for $f$.

| Model | Acc. $f$ | Acc. $g$ | LPIPS ($\downarrow$) | FID ($\downarrow$) | $FF_x$ ($\tau = 0.2$) ($\uparrow$) |
|---|---|---|---|---|---|
| VisCoIN - ResNet50 | 80.56 | 79.44 | **0.545** | 15.85 | **0.146** |
| VisCoIN - ViT-B/16 | 86.66 | **85.86** | 0.582 | **14.88** | 0.081 |

Table 8: Mean and standard deviation for consistency $CC_k$ over all concept functions $\phi_k$ (binary accuracy in %), using $\lambda = 3$. Higher is better. Consistency increases when compared to $\lambda = 2$ (results in main text) and the increase is greatest for VisCoIN.

| Dataset | FLINT | FLAEM | VisCoIN (Ours) |
|---|---|---|---|
| CelebA-HQ | $83.4 \pm 22.9$ | $59.5 \pm 19.2$ | $\mathbf{90.9 \pm 13.8}$ |
| CUB-200 | $76.4 \pm 20.8$ | $56.7 \pm 14.1$ | $\mathbf{93.4 \pm 6.2}$ |
| Stanford Cars | $77.3 \pm 19.6$ | $55.2 \pm 13.5$ | $\mathbf{91.6 \pm 6.3}$ |

# E  ADDITIONAL EVALUATIONS

## E.1  CONSISTENCY WITH HIGHER $\lambda$

We present the results here for evaluating consistency with a higher value of $\lambda = 3$, compared to the main text where $\lambda = 2$. Thus, the constant high activation of $\phi_k(x)$ used to generate "positive" samples of the dataset is increased further. We thus expect the "separation" in the embedding space to increase and consequently a higher performance $CC_k$ of the binary classifier $\varphi_k$ for any $k$. The results for both are presented in Table 8. The results confirm that emphasizing the concept indeed makes the visual modifications more stronger and consistent. Moreover, we also observe that increase in consistency of VisCoIN tends to be larger than increase for other CoIN systems.

## E.2  QUALITATIVE VISUALIZATION FID

While qualitatively, one can clearly observe the difficulty to understand activation maximization based visualization in FLINT. We further support our claim about the unnaturalness of these visualizations compared to visualization in VisCoIN by computing the distance of distributions of visualizations in FLINT, visualizations in VisCoIN and original data distribution, reported in Table 9. Note that we can't use any reconstruction metrics as FLINT visualizations don't reconstruct a given input. Instead they initialize using a given maximum activating sample and execute the optimization procedure of activation maximization to maximally activate a $\phi_k(x)$. We thus compute the FID distance between the visualizations and the data distribution. For VisCoIN visualization we select our most extreme value of $\lambda$. For FLINT visualization we follow their implementation and run the input optimization procedure for 1000 iterations. Since the FLINT visualizations are relatively lot more expensive to compute (1000 backward passes vs 1 forward pass for VisCoIN), we compute the visualizations for 3 maximum activating samples for random 400 relevant class-concept pairs (with $r_{k,c} > 0.5$). Thus the FIDs are computed between 1200 data samples and corresponding visualizations.

Table 9: Quantitative evaluation (FID) of the visualization obtained for interpretation with the original data distribution (1200 samples). Lower is better. For FLINT, visualization is activation maximization output. For VisCoIN, visualization is generated image with $\lambda = 4$.

| Dataset | FLINT | VisCoIN (Ours) |
|---|---|---|
| CelebA-HQ | 21.12 | **9.83** |
| CUB-200 | 26.55 | **11.71** |
| Stanford Cars | 45.72 | **8** |

Table 10: Impact of selecting only the Top-$N$ activated concepts in $\Phi(x)$ before $\Theta$ for prediction, on accuracy of $g$ (in %), for different values of $N$.

| Dataset | $N$ | | | | | | |
|---|---|---|---|---|---|---|---|
| | 4 | 8 | 16 | 32 | 64 | 128 | 256 |
| CUB | 23.75 | 42.25 | 59.28 | 70.07 | 76.25 | 78.97 | 79.44 |
| Stanford Cars | 13.38 | 26.43 | 45.75 | 62.76 | 72.43 | 77.20 | 79.89 |
| CelebA-HQ | 79.92 | 80.63 | 84.00 | 86.90 | 87.71 | – | – |

Table 11: AUC of accuracy curve when gradually adding most activated top-$N$ ($N \in \{2, 4, 8, 16, 32, 64, 128, 200, 256\}$ concepts for each test sample to $\Phi(x)$, along with final accuracy of $g$ (in %) on reconstructed images, on CUB-200 dataset. Higher is better.

| Method | AUC-FF metric | Acc. $g$ on reconstructions |
|---|---|---|
| VisCoIN | **0.407** | **58%** |
| FLINT | 0.042 | 4.5% |

### E.3 Top-N concept activation filter

To preserve maximal amount of interpretable by-design structure, we design the $\Theta$ function as a single linear layer with softmax, similar to other unsupervised CoINs. Nevertheless, this by-design interpretable structure also "erodes" as the size of concept dictionary $K$ increases. For large dictionary sizes, if the activations are not extremely sparse, interpreting the prediction even through a linear $\Theta$ can become tedious and less interpretable (Lipton, 2018).

One way to preserve the interpretable structure even with large $K$ is by controlling the number of non-zero concept activations. We experimented applying a "Top-$N$" function on $\Phi(x)$ before $\Theta$, to keep only the most activated concept for prediction, for different values of $N$, and report results in Table 10. Although it improves interpretability and conciseness of interesting concepts, it comes at the cost of accuracy of the overall system. However, we can see that using about 25% of the most activated concepts still preserves good accuracy in general.

### E.4 AUC-Faithfulness metric

We performed preliminary experiments to compare faithfulness of VisCoIN and FLINT on CUB-200 by computing the area under the curve (AUC) of accuracy w.r.t number of "added" concepts. Specifically, for each test sample $x$, we initialize $\Phi(x)$ to 0, increasingly add the most activated top-$N$ concepts for $N \in \{2, 4, 8, 16, 32, 64, 128, 200, 256\}$ to $\Phi(x)$, plot the accuracy of $g(x')$, $x' = G(\Omega(\Phi(x)_N))$ and compute its AUC.

We report the AUC in Table 11. Note that since the accuracy is on generated images, the accuracy of $g$ is lower than its accuracy on the same dataset. The results are strongly in favor of VisCoIN. We expect VisCoIN to generally outperform other unsupervised CoINs on this metric as it's capable to generate high-quality reconstructions.

### E.5 Additional consistency evaluation

For datasets with annotations for presence/absence of different object parts, consistency for any given concept function $\phi_k$ can also be evaluated by analyzing the "consistency" of their impact on the "presence" of object parts in the generated images. Among the three tasks we evaluate VisCoIN on, the CUB-200 dataset consists of binary annotations for 312 bird-parts for each image. Since such annotations are not available for generated images, prior to the evaluation, we first train a ResNet-50 based predictor to predict probabilities for each bird part. Let this predictor be denoted as $f_{part}$.

For a given concept function $\phi_k$ the goal of this evaluation is to ascertain if there is a bird part which is "consistently affected" when activation $\phi_k(x)$ is modified. To quantify this notion, we first select test images of the classes for which $\phi_k$ is highly relevant, i.e. $r_{k,c} > 0.5$. This set of images is denoted as $\mathcal{X}_k$. For each $x \in \mathcal{X}_k$, we compute $\Phi(x)$ and generate two images with intervened activations, one with $\phi_k(x) = 0$ and other with $\phi_k(x) = \lambda \times \phi_k^{max}$ as done for consistency evaluation

Table 12: Average part consistency over all concepts for FLINT, VisCoIN on CUB-200 (Scale: 0-1, Higher is better).

| Method | Part probability change threshold $\tau$ | | |
| --- | --- | --- | --- |
| | $\tau = 0.05$ | $\tau = 0.1$ | $\tau = 0.2$ |
| VisCoIN | **77.7 ± 13.2** | **64.4 ± 16.1** | **41.7 ± 15.7** |
| FLINT | 56.6 ± 27.5 | 39.6 ± 27.1 | 18.8 ± 20.8 |

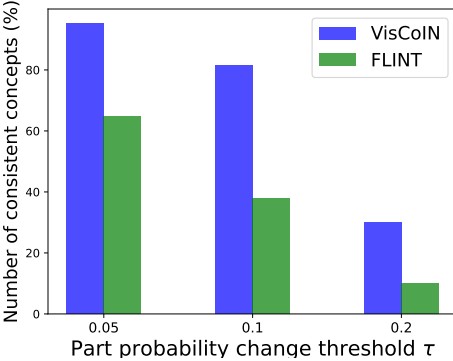

Figure 8: Fraction of concepts which affect some bird part significantly for more than 50% of their relevant images ($CC_k^{part}(\mathcal{X}_k; \tau) > 0.5$). Results reported for different levels of significant part probability change threshold $\tau$ for VisCoIN, FLINT on CUB-200.

in main paper. We use $\lambda = 3$ for this experiment. The generated images are denoted as $x_k^-, x_k^+$ respectively. Note that this dataset preparation is very similar to one used for consistency evaluation in main paper.

We compute part probabilities for all 312 parts as $f_{part}(x_k^-), f_{part}(x_k^+)$. We define $\phi_k$ to *significantly affect* part $j$ for image $x$ if the probability prediction of part $j$ increases when $\phi_k$ activates by more than threshold $\tau$, i.e. $f_{part}(x_k^+)_j - f_{part}(x_k^-)_j > \tau$. For each bird part $j$, its part consistency is computed as fraction of $x \in \mathcal{X}_k$ s.t. $\phi_k$ significantly affects part $j$ in $x$. Consistency of $\phi_k$ is calculated as maximum part consistency over all the parts

$$CC_k^{part}(\mathcal{X}_k; \tau) = \max_j \frac{|\{x : f_{part}(x_k^+)_j - f_{part}(x_k^-)_j > \tau\}|}{|\mathcal{X}_k|}$$

We report results for mean of $CC_k^{part}(\mathcal{X}_k; \tau)$ with $\tau \in \{0.05, 0.1, 0.2\}$ for FLINT and VisCoIN in Table 12. We also report the fraction of concepts that significantly affect some bird part for more than 50% of the images in Fig. 8. Both results indicate that on average concepts in VisCoIN affect some bird part, more consistently, compared to FLINT.

### E.6 SPARSITY EVALUATION

We report in Table 13 sparsity results of VisCoIN and FLINT for CUB-200 at different relevance thresholds. The sparsity is calculated as the average number of relevant concepts per class such that global relevance $r_{k,c} >$ threshold.

FLINT achieves better sparsity because of its use of entropy based losses to compress $\Phi(x)$. While sparsity of VisCoIN could be improved by increasing the $L_1$ regularization weight, we prioritized optimizing for reconstruction/viewability because (i) For previous unsupervised CoINs this is a major limitation, (ii) The current levels of sparsity seemed reasonable (total number of concepts $K = 256$ is much higher than relevant for any class), (iii) Excessive compression of information can make concepts less interpretable and similar to class logits.

Table 13: Sparsity, measured as average number of concepts per class with global relevance higher than a threshold, of VisCoIN and FLINT on CUB-200 dataset. Lower is better.

| Method | Threshold on $r_{k,c}$ | | |
|--------|------|------|------|
|        | 0.7 | 0.5 | 0.2 |
| VisCoIN | 2.3 | 6.1 | 27.1 |
| FLINT | **1.5** | **3.4** | **10.2** |

Table 14: Effect of the weight $\gamma$ of the Reconstruction-Classification loss in the total training loss, measured by Faithfulness, LPIPS and FID, on CUB-200. Faithfulness computed with a threshold of 0.2. **Bold** indicates setting selected for our experiments. Experimentally, $\gamma$ causes tradeoff between faithfulness and quality of reconstruction.

| $\gamma$ | Faithfulness ($\uparrow$) | MSE ($\downarrow$) | LPIPS ($\downarrow$) | FID ($\downarrow$) |
|------|------|------|------|------|
| 0 | 0.001 | 0.142 | 0.52 | 13.11 |
| **0.1** | **0.146** | **0.161** | **0.545** | **15.85** |
| 0.2 | 0.236 | 0.192 | 0.607 | 8.84 |
| 0.5 | 0.24 | 0.209 | 0.634 | 11.54 |

# F   ABLATION STUDIES

We present ablation studies for various components of our system and simultaneously discuss our rationale behind the design selection of these components. Specifically, we study (a) effect of reconstruction-classification loss with weight $\gamma$ in Appendix F.1, (b) role of using a supporting representation $\Phi'(x)$ to assist in reconstruction in Appendix F.2, (c) selection of number of concepts $K$ in Appendix F.3, (d) effect of orthogonality loss in Appendix F.4, (e) effect of fidelity and sparsity loss weights in Appendix F.5, and (f) usefulness of concept translator $\Omega$ in Appendix F.6.

We highlight at this point to the reader that there is an overarching theme that governed many of our design choices. In particular, most of them are based on shaping the systems suitability for better optimization of perceptual similarity for reconstruction. We constantly aim to achieve better reconstruction without major negative impacts for any other properties. This is because for complex datasets (CUB-200, Stanford Cars), the key bottleneck in the design is to achieve the high-quality reconstruction for viewability.

## F.1   RECONSTRUCTION-CLASSIFICATION LOSS

Table 14 reports the perceptual similarity and faithfulness with threshold $\tau = 0.2$ on test data of CUB-200 for different strength $\gamma$. Interestingly, it indicates a tradeoff between faithfulness and perceptual similarity. One possible reason for this tradeoff is that a higher $\gamma$ can push the model to generate more spurious features captured by the classifier at the expense of input quality. Thus, even though the reconstructed images remain relatively far from input this can still lead to "accurate" predictions from the classifier i.e. it predicts the same class as input. Completely removing this loss heavily impacts the faithfulness. However, among the positive $\gamma$, our choice was driven strongly by achieving a reconstruction with high-enough quality and perceptual similarity to enable effective visualization. Thus, we chose $\gamma = 0.1$. The key reason for this is that perceptual similarity is

Table 15: Effect of using the support representation $\Phi'$, measured by MSE, LPIPS and FID, on CUB-200. **Bold** indicates setting selected for our experiments.

| $\Phi'$ | K | MSE ($\downarrow$) | LPIPS ($\downarrow$) | FID ($\downarrow$) |
|------|------|------|------|------|
| **Yes** | **256** | **0.161** | **0.545** | **15.85** |
| No | 512 | 0.178 | 0.568 | 13.52 |
| No | 256 | 0.187 | 0.584 | 9.55 |

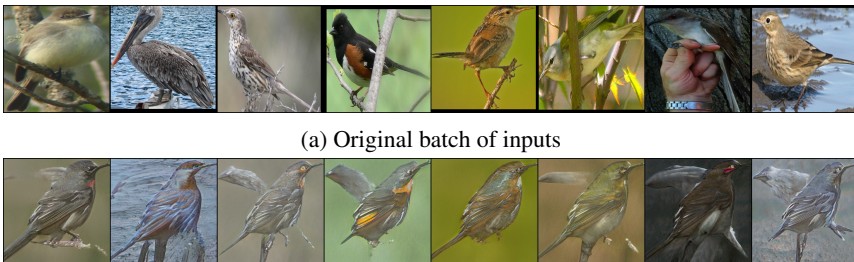

(a) Original batch of inputs

(b) Corresponding reconstructed images

Figure 9: **(a)** Final training batch of images shown to the model. **(b)** Reconstruction obtained using the model trained with $\gamma = 0.2$. Even though FID is better, reconstruction quality is noticeably worse.

Table 16: Impact of the number of concepts $K$ used in $\Phi$, on accuracy of $g$ (in %), LPIPS and FID, for CUB-200. **Bold** indicates setting selected for our experiments.

| K | Accuracy ($\uparrow$) | MSE ($\downarrow$) | LPIPS ($\downarrow$) | FID ($\downarrow$) |
|---|---|---|---|---|
| 512 | 79.78 | 0.156 | 0.537 | 17.27 |
| **256** | **79.44** | **0.161** | **0.545** | **15.85** |
| 128 | 79.03 | 0.183 | 0.578 | 8.64 |
| 64 | 78.91 | 0.203 | 0.624 | 6.3 |

a much better indicator for viewability. For instance, for the $\gamma = 0.2$, even though the FID is better, the reconstruction (LPIPS) is noticeably worse. Reconstruction of the final training batch is indicated in 9, to highlight this issue with high $\gamma$. As is apparent from the figure, the perceptual similarity and consequently the viewability of the model is poor with high $\gamma$. We thus kept a smaller $\gamma$ for CUB-200 and Stanford Cars where the reconstruction is more challenging and slightly higher value for CelebA-HQ.

## F.2    USE OF SUPPORT REPRESENTATION $\Phi'$

The construct of support representation and our use of it is limited to StyleGAN as architecture of $G$. We describe its precise design leveraging the StyleGAN architecture in Appendix C.1.1. While it is not essential to learning and operation of VisCoIN, its inclusion offers a lever to achieve better reconstruction without having to increase the concept dictionary size $K$. Table 15 presents the reconstruction metrics for different concept dictionary sizes and use of $\Phi'$ in reconstruction. Using $K = 256$ with the support representation offers a slightly better reconstruction than even using $K = 512$ but no support representation. As before, we prioritized optimization of LPIPS and using $\Phi'$ assists in achieving better reconstruction whilst allowing us to employ a smaller dictionary.

## F.3    SELECTING NUMBER OF CONCEPTS $K$

The ablation with different number of concepts is given in Table 16 and is an important hyperparameter of the system. While choosing $K$, it is easy to filter out smaller $K$ values as they clearly lead to a worse reconstruction. However, hypothetically a higher $K$ should improve for all the metrics since it allows for more expressivity in the concept representation. Thus finding an upper bound for $K$ is more subjective. Our choice was mainly influenced by (1) the observation that increasing from 256 to 512 offered relatively minimal advantage in reconstruction, and (2) previous methods that used supervised concepts train with 312 concepts. Hence we intended to use a similar dictionary size. Since the Stanford Cars dataset had a comparable number of samples and classes, we used the same number of concepts. For CelebA-HQ, we experimented with reduced $K$ as there are only two classes and the images are less diverse compared to the other two datasets.

Table 17: Impact of orthogonality loss $\mathcal{L}_{orth}$, on accuracy of $g$ (in %), MSE, LPIPS and FID, for CUB-200. **Bold** indicates setting selected for our experiments.

| $\mathcal{L}_{orth}$ | Accuracy ($\uparrow$) | MSE ($\downarrow$) | LPIPS ($\downarrow$) | FID ($\downarrow$) |
|---|---|---|---|---|
| **Yes** | **79.44** | **0.161** | **0.545** | **15.85** |
| No | 79.25 | 0.171 | 0.556 | 9.43 |

Table 18: Effect of weight $\alpha$ on output fidelity loss $\mathcal{L}_{of}$, measured on accuracy of $g$ (in %), LPIPS and FID, for CUB-200. **Bold** indicates setting selected for our experiments. Low $\alpha$ affects performance and high $\alpha$ affects reconstruction quality without much gain in performance

| $\alpha$ | Accuracy ($\uparrow$) | MSE ($\downarrow$) | LPIPS ($\downarrow$) | FID ($\downarrow$) |
|---|---|---|---|---|
| 0.1 | 76.9 | 0.162 | 0.545 | 9.37 |
| **0.5** | **79.44** | **0.161** | **0.545** | **15.85** |
| 2 | 79.63 | 0.187 | 0.586 | 7.58 |

### F.4 ORTHOGONALITY LOSS

For the final layer of $\Psi$, we choose a $1 \times 1$ convolutional layer. Thus, the weights/kernels for this layer can be represented as single matrix of size number of input feature maps times number of concepts $K$. We encourage the $\ell_2$ normalized columns to be orthogonal which in turn encourages each $\phi_k(x)$ to be predicted using different feature maps. We report the quantitative effect of this loss in Table 17. Incorporating this loss offers slight advantage in improved perceptual similarity. However, another key reason we incorporated this loss in our experiments is that we qualitatively observed a greater propensity of multiple concepts highly relevant for a class to capture a common concept about that class. This loss thus offered a way to encourage different concepts to rely on different feature maps. Note that it only affects parameters of final layer of $\Psi$ that outputs $\Phi(x)$, and we do not use any additional hyperparameter for it.

### F.5 OTHER LOSS WEIGTHS

We report the accuracy and reconstruction metrics for different $\alpha$ (weight for output fidelity loss) and $\delta$ (weight for sparsity of activations). A small weight on output fidelity degrades the performance of the system and a high weight affects the reconstruction without benefiting the performance much. We found a balance with $\alpha = 0.5$ which we employed for all datasets. A high $\delta$ impacts both the performance and reconstruction since it encourages activation of smaller number of concepts for any input more strongly. However, eliminating $\delta$ can result in poor sparsity and consequently interpretability of the concept activations for prediction. Our overall strategy was thus to use a high enough $\delta$ that it does not significantly affect reconstruction quality. For CUB-200 and Stanford Cars, due to a greater need of prioritizing reconstruction we opted for a smaller $\delta = 0.2$, while for CelebA-HQ, since obtaining a good reconstruction was relatively easier, we opted for a higher $\delta = 2$.

We keep a fixed $\beta = 3$ weight for LPIPS reconstruction loss throughout, for all our datasets and ablations. Even though the system still provides meaningful results for $\beta < 3$, the lower values were ruled out mainly because of the importance of a lower perceptual similarity loss, mentioned earlier. The higher values were ruled out because in our initial experiments we observed some instability with high $\beta > 4$. Thus we fixed $\beta = 3$ for all datasets which provided a good balance.

### F.6 BASELINE FOR CONCEPT TRANSLATOR $\Omega$

We compare our system with a variant where we directly use $\Phi(x)$ as latent vector $w_x$ for $G$, eliminating $\Omega$. While our model allows this design, it comes with certain limitations: (i) The user can't control the number of concepts. They are forced to employ a concept dictionary of same size as dimension of the latent space. (ii) Since the generator is pretrained and fixed, the resulting $\Phi(x)$ learnt is not sparse. (iii) Finally, in particular for GANs, it forcibly associates concept functions

Table 19: Effect of weight $\delta$ on sparsity, measured on accuracy of $g$ (in %), MSE, LPIPS and FID, for CUB-200. **Bold** indicates setting selected for our experiments. A high $\delta$ can affect both reconstruction and performance.

| $\delta$ | Accuracy ($\uparrow$) | MSE ($\downarrow$) | LPIPS ($\downarrow$) | FID ($\downarrow$) |
|---|---|---|---|---|
| **0.2** | **79.44** | **0.161** | **0.545** | **15.85** |
| 2 | 79.54 | 0.174 | 0.562 | 11.44 |
| 20 | 76 | 0.201 | 0.629 | 9.83 |

Table 20: Impact of concept translator $\Omega$, on accuracy of $g$ (in %), MSE, LPIPS and FID, for CUB-200. **Bold** indicates setting selected for our experiments.

| $\Omega$ | Accuracy ($\uparrow$) | MSE ($\downarrow$) | LPIPS ($\downarrow$) | FID ($\downarrow$) |
|---|---|---|---|---|
| **Yes** | **79.44** | **0.161** | **0.545** | **15.85** |
| No | 79.24 | 0.182 | 0.572 | 15.96 |

with columns of identity matrix as directions in latent space. Using an $\Omega$ (for instance a linear layer) allows the model to learn general directions in the latent space to associate to each concept function, which aligns with the conventional strategy for latent traversal inside GAN. As can be seen in Table 20, removing $\Omega$ leads to poorer reconstruction.

### F.7 OUTPUT FIDELITY LOSS

One can consider using cross-entropy loss with ground truth labels instead of "output fidelity loss". We specify the possibility to use both when describing the general architecture of unsupervised CoINs (in Section 3.1). We decided to use the output fidelity loss following Sarkar et al. (2022). The current design also draws inspiration from knowledge distillation setting, in which a student model is trained to reproduce output of a teacher model. One additional perk of using this "output fidelity loss" during VisCoIN training is that it can also be applied for images without annotations, such as images sampled from $G$ (further details about VisCoIN training in Appendix C). This provides additional guidance and stability to train $g$. For completeness, we include in Table 21 an experiment of VisCoIN trained using a standard cross-entropy loss with ground truth labels on CUB-200 dataset.

Table 21: Comparison between the "Output fidelity loss" and "Cross-entropy loss", measured by accuracy of interpreter $g$ (in %), reconstruction quality (LPIPS), faithfulness (median $FF_x$ for threshold $\tau = 0.2$), on CUB dataset.

| Model | Acc. $g$ ($\uparrow$) | LPIPS ($\downarrow$) | $FF_x$ ($\tau = 0.2$) ($\uparrow$) |
|---|---|---|---|
| VisCoIN - Output fidelity loss | **79.44** | **0.545** | **0.146** |
| VisCoIN - Cross-entropy loss | 78.89 | 0.559 | 0.076 |

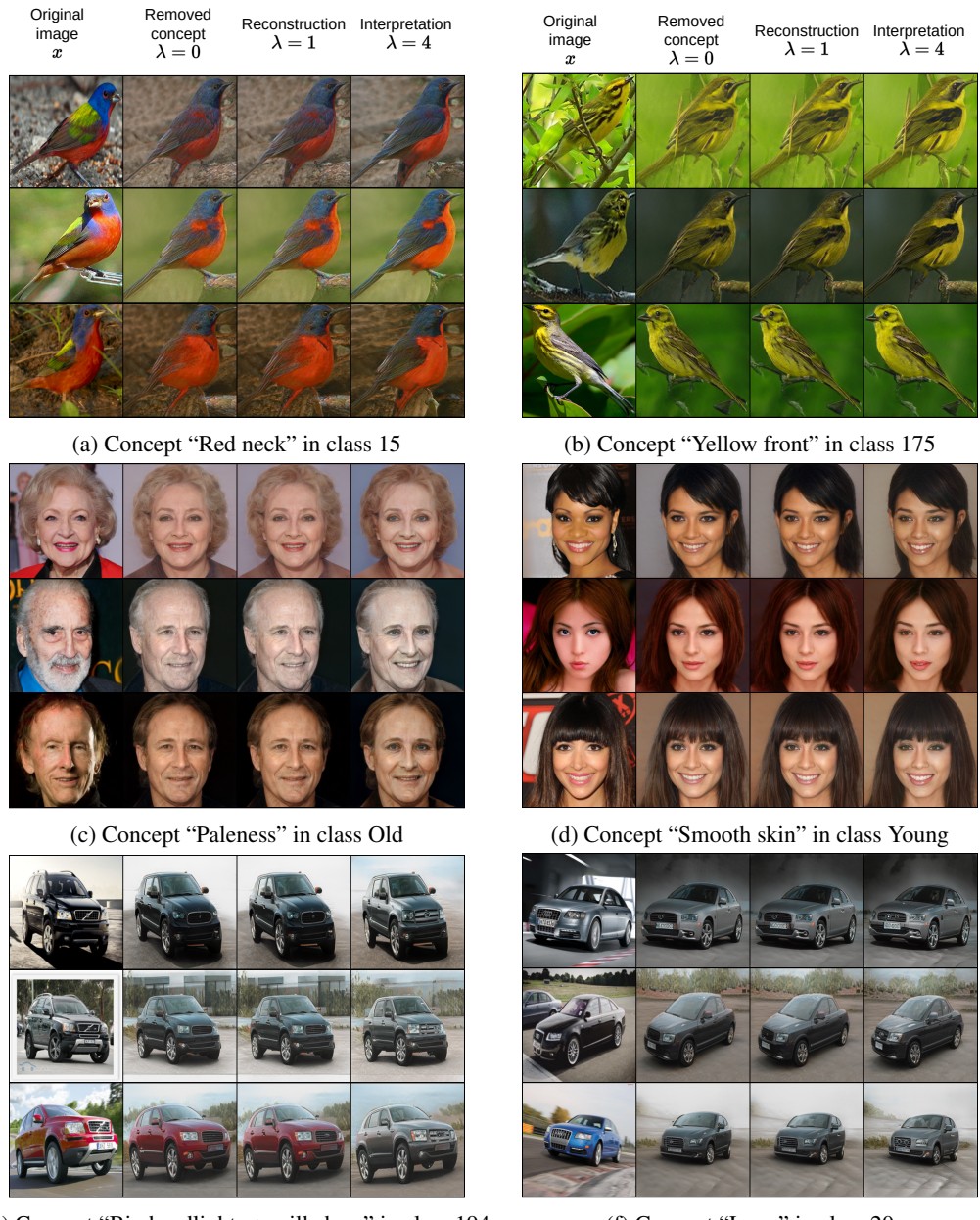

Figure 10: Additional qualitative examples obtained for different concepts, classes on **(a)-(b)** CUB-200, **(c)-(d)** CelebA-HQ, **(e)-(f)** Stanford-Cars datasets. On each subfigure, first column corresponds to maximum activated samples $x$ for class-concept pairs with high relevance ($r_{k,c} > 0.5$), third column to reconstructed image obtained with original $\Phi(x)$, while second and fourth columns to the images obtained by imputing respectively $\phi_k(x) = 0$ and $4 \times \phi_k(x)$ in $\Phi(x)$.

## G ADDITIONAL ANALYSIS

We show additional visualizations for different highly relevant class-concept pairs ($r_{k,c} > 0.5$) in Fig. 10. For each class-concept pair, we show the effect of modifying the concept activation on the generated output for three maximum activating training samples. For each sample (on the far-left), we show the corresponding generated outputs for $\lambda = 0$ (center-left), $\lambda = 1$ (center-right) and $\lambda = 4$ (far-right).

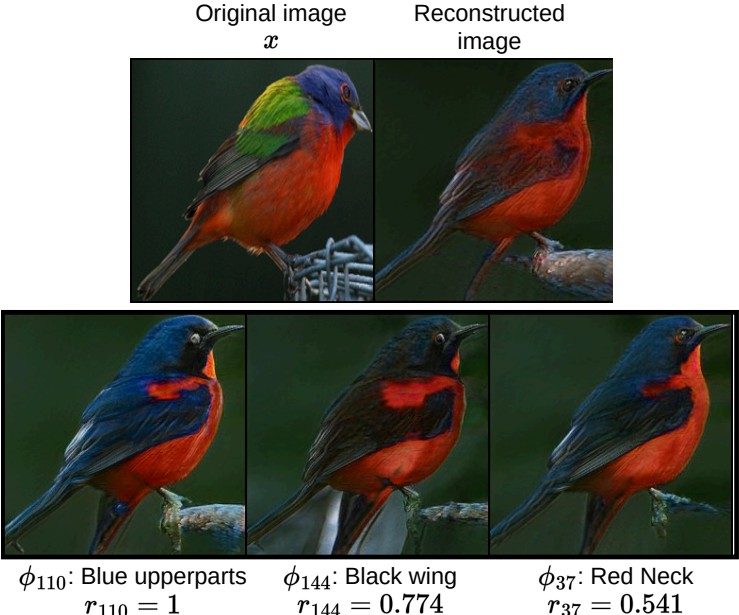

Figure 11: Local interpretation on CUB dataset (sample from class 15) for different concepts along with their relevance scores.

## G.1 LOCAL INTERPRETATIONS

We present in Fig. 11 local interpretations on test samples of CUB dataset, for different concepts relevant to that class. For a single test sample, a user obtains multiple concepts that are relevant for the prediction of the test sample. We extract the concepts with high relevances for the given input. For each concept one can use the visualization pipeline introduced to visualize what part of the input activates the respective concept. Note that the visualizations for $\phi_{110}, \phi_{37}$ remain consistent with their global visualizations.

## G.2 USE OF DIFFERENCE IMAGES CAN BE USEFUL

To better highlight regions in the image impacted by modifying concept activations, one can additionally visualize the *differences* between two generated outputs. We show visualizations with the difference in the generated outputs in Fig. 12. Again, we selected highly relevant class-concept pairs ($r_{k,c} > 0.5$), and show the effect of modifying the concept activation on the generated output for three maximum activating training samples. For each sample (on the far-left), we show the corresponding generated outputs for $\lambda = 1$, $\tilde{x}$ (center-left), and $\lambda = 4$, $\tilde{x}'$ (center-right). We then compute and show the difference between the two generated outputs $\tilde{x}' - \tilde{x}$ (far-right). It is worth noting that this exact strategy might not work as effectively for all types of concepts and can require modifications. For example, if "black feathers" are emphasized by a concept, the increasing "black" color won't be visible in the difference between $\tilde{x}', \tilde{x}$. Instead, one could either visualize the reverse difference between $\tilde{x}, \tilde{x}'$ to identify a color being "removed" or visualize the energy of difference for each pixel to identify which regions are modified the most.

## G.3 AVERAGE LATENT VECTOR

We illustrate through example on CUB in Fig. 13 that the average latent vector of pretrained $G$ is typically representative of the dataset.

| Original image $x$ | Reconstruction $\lambda = 1$ | Interpretation $\lambda = 4$ | Difference of images |
|---|---|---|---|

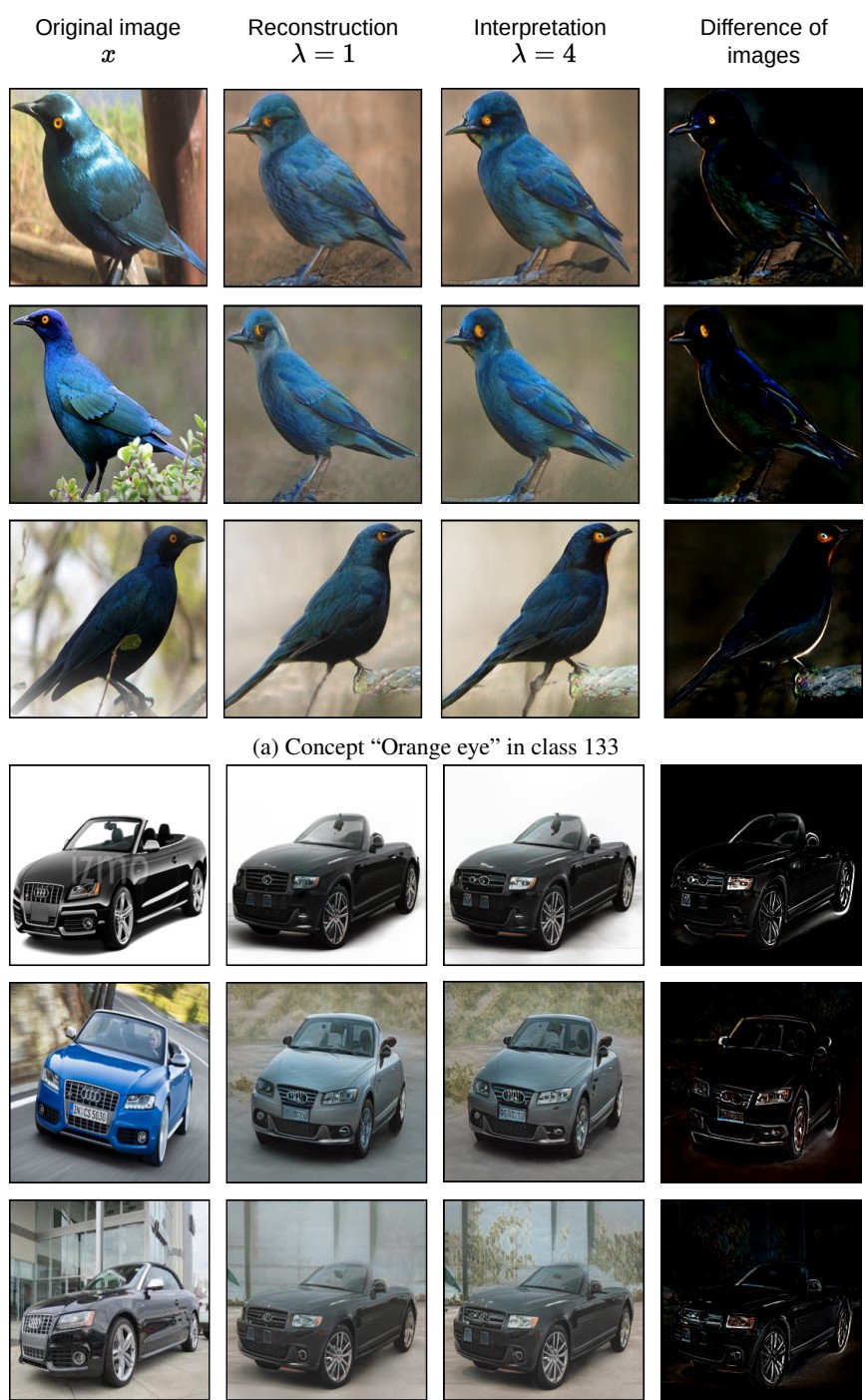

(a) Concept "Orange eye" in class 133

(b) Concept "Headlight shine" in class 18

Figure 12: More qualitative examples obtained for different concepts, classes on **(a)** CUB-200, **(b)** Stanford-Cars datasets. On each subfigure, first column corresponds to maximum activated samples $x$ for class-concept pairs with high relevance ($r_{k,c} > 0.5$), second column to reconstructed image obtained with original $\Phi(x)$, third column to the image obtained by imputing $4 \times \phi_k(x)$ in $\Phi(x)$, and fourth column shows *the difference between third and second images*.

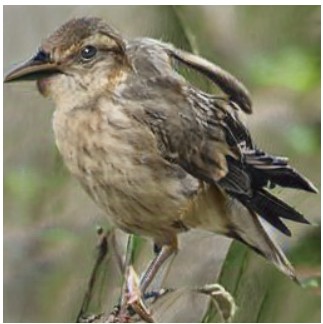

Figure 13: Generated image for average latent vector ("center" of latent space) in CUB (temporary figure).

## H    LIMITATIONS OF APPROACH

- As is the case for other CoINs learning unsupervised concepts, the proposed system cannot *guarantee* that concepts precisely correspond to human concepts and not encode other additional information. However, one interesting aspect is that visualization process in VisCoIN gives a better handle at identifying any deviations as visualizations in other unsupervised CoINs can be much harder to understand with granularity for large-scale images.
- The choice of using a pretrained $G$ improves training time, complexity and reusability, but also implies that the system's quality is limited by the quality of the pretrained $G$. For instance, for visualization, if $G$ can't generate some specific feature, it can be difficult to visualize a concept $\phi_k$ that encodes that feature.
- For the case of single FC layers in $\Omega$, our visualization process follows linear trajectories in the latent space of $G$ when modifying an activation. Recent work has shown that linear trajectories are not necessarily optimal for latent traversals (Song et al., 2023).

## I    POTENTIAL NEGATIVE IMPACTS

Given that the understanding of neural network decisions is considered as a vital feature for many applications employing these models, specially in critical decision making domains, we expect our method to have an overall positive societal impact. However, in the wrong hands almost any technology can be misused. In the context of VisCoIN, it can be used to provide deceiving interpretations by corrupting its training mechanisms (for example by training on misleading annotated samples, using deliberately altered pretrained models etc.). Thus, we expect a responsible use of the proposed methodology to realize its positive impact.

