# OpenReview forum: "Restyling Unsupervised Concept Based Interpretable Networks with Generative Models"
_ICLR.cc/2025/Conference — ICLR 2025 Poster_

### Official Review · Reviewer_Lumc · 2024-10-31

**Soundness:** 3
**Presentation:** 3
**Contribution:** 3
**Rating:** 6
**Confidence:** 4

**Summary:**

The authors propose a method to map concept features to a generative model latent space to have a better interpretation of the learned concepts. They focus on concept-based interpretable networks (COIN) to build the visualization system. They evaluate the interpretation in terms of the accuracy of the prediction network (fidelity to output), the fidelity of the reconstruction (fidelity to input), and the fidelity and consistency of the learned concepts. They also propose the analysis of sparsity and viewability (capacity of reconstruction over concepts).

**Strengths:**

- Writing and Implementation Details: I appreciate the authors' attention to motivating and well-describing the methodology, including detailed implementation parameters in the main and supplementary texts.

- Improvement of general interpretability: the authors present an interesting approach to improve the interpretability of visual networks by learning sparse concepts that can be quantified and visualized.

- Numerous evaluation methods: an important aspect of this work is the aspect in which the methodology is evaluated, which configures an important validation of the proposition: prediction accuracy, fidelity to reconstruction, faithfulness, consistency, sparsity, and viewability.

**Weaknesses:**

- Combination of two complex models to interpret and explain: Have you analyzed the problems to include the pre-trained generative network? Is there a bias that could change the visualizations even with the same feature extractor? It seems to me that you are including a new possible bias in the pipeline.

    As I understand it, the concept functions are learned along with the concept translator, but in doing so, the latent space of the generator affects the learning concept functions. Can this mean that the bias from both networks is present in these concepts? Are we explaining the feature extractor or the generative model?

- The number of concepts and their semantics: How was the size of the concept dictionary chosen? What happens with smaller K? You show some ablation studies, and then you decide on 64. What about smaller numbers of concepts? It would be interesting to see how the visualizations change depending on that number.

- Faithfulness and consistency: In the paper, the authors mention these evaluations as new, but faithfulness is commonly used as a technique of evaluation [1]. I also understand the approach used as feature (concept) removal. A suggestion to the authors is to also evaluate the effect of inserting only the top concepts into the generated image. Moreover, an interesting evaluation would be the one proposed in the paper of XRAI [2], to iteratively insert the top concepts and analyze the AUC of the accuracy curve. Regarding the consistency, I like this idea, I just have questions about the classifier used to separate the representations with and without the concepts, what is its architecture? Is it linear before decision? How many layers? Also, I would suggest an additional experiment: use a part label dataset like CUB-200 with bird parts and check if the most changed parts when removing a concept are the same for birds in the same class.

- Sparsity and Viewability: I don't see much discussion of these metrics other than the loss used during training. I would like to see at least some viewability analysis (qualitative) and possibly a human evaluation compared to the baseline approaches.

[1] Covert, I., Lundberg, S., & Lee, S. I. (2021). Explaining by  removing: A unified framework for model explanation. Journal of Machine  Learning Research, 22(209), 1-90.

[2] Kapishnikov A., Bolukbasi T., Vi'egas F., Terry M. XRAI: Better Attributions Through Regions. ICCV. 2019.

**Questions:**

Some questions are already presented in the weakness section.

Other questions:

→ Why does the bird head of the first row in Figure 4 also increase in size with the red eye? Is this related to another concept?

→ How did you determine the "name" of the concept? Is it a deduction?

**Details Of Ethics Concerns:**

No concerns.

---

> ### Author Response · Authors · 2024-11-21
> **Response to Reviewer Lumc (1/2)**
>
> Thank you for the review. We respond to your comments pointwise below:
> ### Weaknesses
>
> * **W1:** These are interesting points and related to the **second limitation in Appendix H**. Our visualization is limited by the quality of the generative model and its latent space. If a generator is not capable of representing a feature in its latent space, then we can't visualize it.
>
>     An implicit assumption is that the generator is good at modelling input related features. If true, it is more reasonable to expect that the generator provides a rich latent space that represents a larger range of input features than needed for classification. In such a case the generator shouldn't induce additional bias for learning concepts than that of the classifier. The model will learn to map the concepts extracted by the classifier to latent space of the generator. This is why we require the generative model to be good at generating the given data distribution and finetune our $G$ before training VisCoIN when that is not the case.
>
> * **W2:** We analyzed the impact of K in Appendix F.3. Smaller K leads to worse reconstruction (and accuracy), while higher K improves all metrics, since it allows for more expressivity in the concept representation. However, finding a good upper bound is subjective. We want a small dictionary for reduced overhead when interpreting the concepts, but a higher dictionary can improve disentanglement and reconstruction. In practice, we selected K=256 for CUB and Stanford Cars, and K=64 for CelebA. Our choice was influenced by (i) the number of classes, (ii) using number of concepts in supervised CoINs as a reference, and (iii) experiments showing that increasing K did not noticeably improve results.
>
> * **W3:**
>     - **Evaluation novelty:** We would like to clarify that we propose a novel **strategy** to evaluate faithfulness **in the context of unsupervised CoINs**, that uses the decoder/generator to explicitly modify the semantic content encoded by the concepts. We agree faithfulness metric is widely used for interpretability methods. In the main text too we initiate discussion about our faithfulness evaluation by recognizing the previous strategies to evaluate faithfulness (L380--384). We'll modify the text to make it explicit everywhere.
>
>     - **Details about consistency**: We cover the details in Appendix C.3.1. We trained a linear SVM from the output of the second block of the ResNet-50 encoder, which are feature vectors of dimension 512 after pooling each feature map. We are currently implementing the additional consistency experiment, and hope to complete it within the discussion period, but it will be added in the revised version.
>
>     - **AUC metric for faithfulness**: Thanks for the nice suggestion. We performed preliminary experiments comparing VisCoIN and FLINT on CUB-200 by adding most activated $N=4, 64, 128, 256 \text{(all)}$ concepts for each test sample to $\Phi(x)$ initalized to 0, and plotting the accuracy of g(.). We report the AUC below. Note that since the accuracy is on generated images, it is lower than accuracy of $g(.)$ on dataset.
>     Method | AUC-FF metric | Accuracy of $g(.)$ on full reconstructions
>     --- | --- | ---
>     VisCoIN | 0.396 | 58%
>     FLINT | 0.041 | 4.5%
>
>         We'll repeat the experiments with greater resolution for $N$, but the results are already strongly in favor of VisCoIN. We expect VisCoIN to generally outperform other unsupervised CoINs on this metric as it's capable to generate high-quality reconstructions.
>
> * **W4:** Since the key aspect of the viewability property is to enable high-quality reconstruction of $x$ from $\Phi(x)$ (L247, Sec 3.2), the viewability is quantitatively assessed by the reconstruction metrics, particularly LPIPS and FID. VisCoIN generally significantly outperforms the baselines thanks to the powerful pretrained generator. We'll make it explicit in Sec 4.1 to indicate "Fidelity of Reconstruction" quantifies viewability.
>
>     We report below sparsity results of VisCoIN, FLINT for CUB at different relevance thresholds. The sparsity is calculated as the average number of relevant concepts per class such that global relevance $r_{k, c} > \text{threshold}$
>     Method | threshold = 0.7 | threshold = 0.5 | threshold = 0.2
>     --- | --- | --- | ---
>     VisCoIN | 2.3 | 6.1 | 27.1
>     FLINT | 1.5 | 3.4 | 10.2
>
>     FLINT achieves better sparsity because of its use of entropy based losses to compress $\Phi(x)$. While sparsity of VisCoIN could be increased by increasing the l1 regularization weight, we prioritized optimizing for reconstruction/viewability because (i) For previous unsupervised CoINs this is a major limitation, (ii) The current levels of sparsity seemed reasonable (total number of concepts K = 256 is much higher than relevant for any class), (iii) Excessive compression of information can make concepts less interpretable and similar to class logits. For completeness we will add this discussion in appendix.

---

> ### Author Response · Authors · 2024-11-21
> **Response to Reviewer Lumc (2/2)**
>
> ### Questions
>
> * **Q1:** The concepts do not always precisely correspond to what we would consider as concepts from a human perspective. Some of these concepts can be entangled, especially if they are attending the same region of the image or object. We try to disentangle them through constraints like the orthogonality loss, but it is difficult to completely eliminate, even for supervised CoINs [A]. We discuss this limitation in Appendix H.
>
> [A] M. Havasi, S. Parbhoo, F. Doshi-Velez. "Addressing Leakage in Concept Bottleneck Models". NeurIPS 2022
>
> * **Q2:** We manually inferred the "name" of concepts from the changes that we observed. This is commonly done for naming visual concepts, e.g. with most activating samples (MAS) or in concept discovery (CRAFT [B], ACE [C], etc).
>
> [B] T. Fel, et al. "Craft: Concept recursive activation factorization for explainability." CVPR 2023
> [C] A. Ghorbani, et al. "Towards automatic concept-based explanations." NeurIPS 2019

---

> > ### Comment · Reviewer_Lumc · 2024-11-25
> >
> > Thank you for the clarifications and for considering the suggestions. I will keep my score.

---

### Official Review · Reviewer_2hPr · 2024-11-02

**Soundness:** 3
**Presentation:** 3
**Contribution:** 3
**Rating:** 6
**Confidence:** 3

**Summary:**

The paper proposes a new method of interpreting latent features which builds upon concept-interpretable neural networks (CoINs) and adds the new means of interpretation through mapping the latent features into the space of generative models. It helps address the known problem of interpretations of concepts-based models in a way that is complementary to the existing explanations, which often relates the concepts to the real data.

**Strengths:**

- Correctness: I checked the notation and did’t see any errors.
- Reproducibility: the paper looks reproducible to me (see Question 3 below)
- Novelty: the method builds upon the existing CoINs methods but provides the new means of interpretation comparing to the original CoIN methods, and therefore is novel in this way.
- Clarity: the outline of the paper is clear, however with some suggestions on the presentation
- Significance: this work complements previous work on concept-based interpretation. The approach is well-motivated by the need in providing more detailed interpretations of the vision recognition models and is, as the authors discuss in the introduction, well-grounded in the state-of-the-art. The significance of the work is mostly empirical, with the authors presenting both the advantages of the method and the evaluation protocol.

**Weaknesses:**

- Clarity: There are a few questions below which could help clarify upon the relation between the method and the by-design approaches.

**Questions:**

1. It would be important to clarify upon the limitations regarding to the relation of the proposed method to the inherently-interpretable class of methods. Such methods are supposed to provide by-design explanation, where the output is causally related to the explanation. For example, concept bottleneck models (Koh et al, 2020),  ensure such by-design claim by making the prediction directly rely upon the intermediate concepts (i.e., first, we predict a  number of properties, and then infer the class solely on these properties).  In this work, I can identify two points where this by-design property could be broken: (1) I understand that the output depends upon the whole set of features, which means that the less-contributing features can still influence the prediction enough to change the label. Alternatively, one can select only a part of features, perhaps at a cost of accuracy (2) the matching between the latent space and the generative model is performed using a concept translator; it means that in some scenarios the mapping between the latent space and the generative model can be imperfect. One might find it useful, perhaps even at a cost of accuracy, to bridge these gaps and make the classification inherently-interpretable. It may be achieved, to address the by-design limitation (1), by only performing the prediction from the features which contribute the most and discard the rest, and provide by-feature explanations for these features. To address the limitation (2), one may think of learning the latent space in a way that it coincides with the generative model’s one (i.e., through distillation). I wonder if this model allows for this?
2. I see that the model is evaluated using ResNet-50 backbone. I wonder if it can generalise to the transformer-based architectures? In relation to this, would the authors clarify upon the computational overhead of the proposed method in comparison with the CoIN and the ResNet-based models?
3. Figure 4 states “Visual modifications of more local concepts indicated by red boxes” I am not sure I could  understand what it would mean and entirely follow how the red box visualisation works .

---

> ### Author Response · Authors · 2024-11-21
> **Response to Reviewer 2hPr**
>
> Thank you for the review. We respond to your questions pointwise below:
>
> ### Questions
>
> * **Q1:** In our system design, we kept architectures taking $\Phi(x)$ as input as simple as possible to maximally preserve interpretability, i.e. $\Omega$ is a single linear layer and $\Theta$ is also a single linear layer with softmax. Although, we agree that even with a linear layer, interpretability for prediction can erode if the size of concept dictionary is too large and activations are not sparse. This is a limitation in general for all CoINs (supervised & unsupervised).
>     - (1) We experimented applying a "Top-N" function on $\Phi(x)$ before $\Theta$, to keep only the most activated concept for prediction, for different values of N, and report results below. As mentioned by the reviewer, although it improves interpretability and conciseness of interesting concepts, it comes at the cost of accuracy of the overall system. However, we can see that using about 25% of the most activated concepts still preserves good accuracy in general.
> Dataset | N=4 | N=8 | N=16 | N=32 | N=64 | N=128 | N=256
> --- | --- | --- | --- | --- | --- | --- | ---
> CUB | 23.75 | 42.25 | 59.28 | 70.07 | 76.25 | 78.97 | 79.44
> Stanford Cars | 13.38 | 26.43 | 45.75 | 62.76 | 72.43 | 77.20 | 79.89
> CelebA-HQ | 79.92 | 80.63 | 84.00 | 86.90 | 87.71 | x | x
>
>     - (2) We include in Appendix F.6, an experiment where we directly use $\Phi(x)$ as latent vector $w_x$ for G, eliminating $\Omega$. While our model allows this design, it comes with certain limitations: (i) The user can't control the number of concepts. They are forced to employ a concept dictionary of same size as dimension of the latent space. (ii) Since the generator is pretrained and fixed, the resulting $\Phi(x)$ learnt is not sparse. (iii) Finally, in particular for GANs, it forcibly associates concept functions with columns of identity matrix as directions in latent space. Using an $\Omega$ (for instance a linear layer) allows the model to learn general directions in the latent space to associate to each concept function, which aligns with the conventional strategy for latent traversal inside GAN.
>
>
> * **Q2:**
>     - We are happy to report an additional experiment using a ViT-B/16 for $f$ on CUB dataset, while keeping other architectures almost identical. We started from a pretrained ViT-B/16 and only finetuned the classification head on CUB, to use as $f$. We take the patch embeddings of final layer as input to $\Psi$. We keep identical $\Theta, \Omega$, hyperparameters, and slightly modify $\Psi$ for reduced number of feature maps. As can be seen from the numerical results below, we achieve better accuracy thanks to a better pretrained $f$, but reconstruction and faithfulness are worse. The results could be improved by better desigining $\Psi$ and accessing more internals embeddings. However, they certainly show that VisCoIN can generalize to other backbone architectures.
> Model | Acc. f | Acc. g | LPIPS ($\downarrow$) | FF ($\tau = 0.2$) ($\uparrow$)
> --- | --- | --- | --- | ---
> VisCoIN - ResNet50 | 80.56 | 79.44 | **0.545** | **0.146**
> VisCoIN - ViT-B/16 | 86.66 | 85.86 | 0.582 | 0.081
>
>     - Computational overhead of VisCoIN vs other unsupervised CoINs:
>         * **Number of trainable parameters**: The subnetworks $\Psi, \Theta, \Omega$ are very light compared to $f, G$. The number of parameters is thus comparable or fewer than other unsupervised CoINs since both $f$ and $G$ are pretrained and fixed.
>         * **Training VisCoIN**: VisCoIN is 2-4 times slower to train than other unsupervised CoINs because of the passes through the generator. However, we are still able to train VisCoIN on each task in a single day on a single GPU thanks to the use of pretrained $f, G$.
>         * **Inference, Interpretation time**: The inference time is the same as other CoINs as it only uses $f, \Psi, \Theta$ to compute $g(.)$, and interpretation is similar or faster than unsupervised CoINs, since no input optimization is required.
>
> * **Q3:** We manually added red boxes to indicate the main regions where the modification is occurring in generated images. We will improve the figure caption and hope that it is more clear.

---

> > ### Comment · Reviewer_2hPr · 2024-11-22
> >
> > Many thanks, I've checked the responses to my review and to the other reviewers. This answers my questions in general, and I hope the authors could implement these changes.
> >
> >  I think the experiment for Q1 addresses the concern about the prediction from Top-N concepts, and it would be great to see it in the updated version. It might be also good to refer, perhaps in the conclusion or the limitations, to this trade-off between interpretability and accuracy when selecting the number of concepts to perform the prediction.

---

### Official Review · Reviewer_9qiE · 2024-11-03

**Soundness:** 3
**Presentation:** 3
**Contribution:** 3
**Rating:** 8
**Confidence:** 3

**Summary:**

This paper proposes a new system for concept-based interpretable networks, where the learned concepts are transformed to the latent space of a generative model. This transformation is learned, enforcing consistency by minimising the distance loss between the original image, and the image produced by the generative model using the concepts of the original image. This allows users to easily visualise the learned concepts, by generating images in a range for a particular concept.

**Strengths:**

- I agree with the idea that using examples from your training set that maximally activate a certain concept is not ideal, and the authors address this nicely. The concept visualisations generated are much simpler to understand than concept attribution maps and visualising image patches that maximise the concept.
- Leverages pretrained generative models rather than training an additional decoder, instead using a less complex concept translator, making the method more efficient.
- The measure of consistency of concept visualisation is novel and seems appropriate as a proxy for human experiments.

**Weaknesses:**

- Some of the reconstructions shown throughout the paper are quite far from the original image, and sometimes changing the image quite drastically, such as Figure 4b and more examples in the appendix. This is one benefit existing methods like concept attribution maps have, the concepts are highlighted on the original images rather than relying on a reconstruction.
- Some details in C1.1 on how reconstruction is performed with StyleGAN are important, for example the need for using a secondary, unconstrained representation $\Phi'(x)$, should be in the main text, as this suggests that reconstruction purely based off the interpretable concepts is not possible. This could be added in Section 3.2.

**Questions:**

- The authors state that the maximum $\lambda$ that is reliable is 3 or 4. Is this due to the magnitude of the concept seen by the translator network during training? And if so, couldn't think potentially be an issue when setting $\phi_k(x)$ to 0 when measuring faithfulness, as this might be outside of the range seen during training for a particular concept.
- $\Omega$ used is a very simple architecture, did the authors ever experiment using multiple layers to allow a more complex transformation?

---

> ### Author Response · Authors · 2024-11-21
> **Response to Reviewer 9qiE**
>
> Thanks for the review. We address the concerns pointwise below:
>
> ### Weaknesses
>
> * **W1:** Compared to other unsupervised CoIN, we make significant advancements in reconstruction, as assessed by the different reconstruction metrics, but we agree it is not perfect.
> While concept attribution maps are very useful to highlight regions of relevance for a concept, similar to feature attribution approaches, they are not as effective in revealing the "semantic content" detected by a concept.
> Our intervention and visualization approach for interpretability is aimed at filling this gap. Our approach can be useful even if the reconstruction is not perfect, provided it's close enough so that changes in it can be grounded to the original image. Most importantly though, **we do not believe these two are in competition with each other but complementary**. We raised the point about using a tool to highlight relevant regions for a concept in Appendix G.2 such as the difference of images to assist in localizing the modifications.
>
> * **W2:** The unconstrained supporting representation leverages specificities of the StyleGAN family of architectures. While it improves reconstruction quality and results, it is not crucial to the understanding and proper learning of the overall system, and we thought it would hinder clarity in the main text. We show in Appendix D that the system provides meaningful results with experiments using ProgressiveGAN and $\beta$-VAE that do not rely on $\Phi^\prime$. Nonetheless, we will add a reference in Section 3.2 to the detailed discussion in the Appendix.
>
> ### Questions
>
> * **Q1**: We think that the constraint on $\lambda$ comes from the limitation of doing linear traversal in the latent space of G. An unreasonably high $\lambda$ can push the resulting latent vector outside the image manifold relevant to the dataset. Setting $\phi_k(x)=0$ generally should not cause problems as it corresponds to moving towards average latent vector of the generator, which is generally the "centre" of the latent space. Also, since only a fraction of concept dictionary is generally relevant for any class (more details in our response W5 to Reviewer Lumc), any given $\phi_k(x)$ is frequently 0 for many input samples.
>
> * **Q2**:  We only considered designing $\Omega$ with single fully connected layers for experiments, such that it associates each concept function $\phi_k$ to a linear direction in latent space of $G$. Intervening on concept activation $\phi_k(x)$ corresponds to linear traversal in latent space, which is a common practice for latent traversal in generative models. An arbitrary complex transformation would result in reduced interpretability of image transformations.
>
>     However, it is an intriguing suggestion, as recent work has shown that linear trajectories are not always optimal for latent traversal, which. We discuss this as a limitation in Appendix H. More complex design of $\Omega$ could achieve meaningful non-linear traversal and improve reconstruction. We leave it as a future work since its a challenging problem in itself.

---

> > ### Comment · Reviewer_9qiE · 2024-11-25
> > **Reviewer 9qiE Response**
> >
> > > Our approach can be useful even if the reconstruction is not perfect, provided it's close enough so that changes in it can be grounded to the original image.
> >
> > I agree that the method is useful as long as the reconstruction is close enough.
> >
> > > We raised the point about using a tool to highlight relevant regions for a concept in Appendix G.2 such as the difference of images to assist in localizing the modifications.
> >
> > I think this is a different issue, the difference between images generated with different $\phi_k(x)$ is interesting, but my comment was regarding the difference between the original and reconstructed image.
> >
> > > Setting generally should not cause problems as it corresponds to moving towards average latent vector of the generator, which is generally the "center" of the latent space.
> >
> > Just out of interest, have you had a look at what image the "center" of the latent space generates?
> >
> > I have had a look at the other reviews and authors response and adjusted my score.

---

> > > ### Author Response · Authors · 2024-11-27
> > >
> > > Thank you for the positive update!
> > >
> > > The center of latent space during pretraining of G moves towards some common features in the dataset. The generated images tend to look representative of the underlying data distribution. For eg. in case of CUB, our pretrained G generates a small bird with brown color, white front with brown spots and brown-black wings/feathers. Image is attached in revised version (Fig. 13 at the end of appendix).
> > >
> > > To clarify, we raised the use of "difference between images", as a tool to highlight relevant changes on *reconstructed images*, similarly to how concept attribution maps highlight relevant region for concept activation.

---

### Official Review · Reviewer_gc79 · 2024-11-03

**Soundness:** 3
**Presentation:** 3
**Contribution:** 3
**Rating:** 6
**Confidence:** 3

**Summary:**

A primary limitation in human-understandable concept learning within intrinsic XAI approaches is the effective visualization of learned, unsupervised concept dictionaries, particularly for large-scale images. To address this challenge, the authors introduce VisCoIN, a novel, concept-based interpretable network that includes a concept translator, which maps concept vectors into the learned space within a generative model. Experimental results visualizing the learned concepts highlight the efficacy and practical value of the proposed approach.

**Strengths:**

- S1: The definition of viewability and the proposed method are clear, well-founded, and intuitive. Experimental visualizations of the learned concept effectively demonstrate its amplified appearance as the lambda value varies within the concept function, providing compelling evidence of the approach's effectiveness.

**Weaknesses:**

- W1: The authors introduced an additional pretrained classifier f, and incorporated its output along with g(x) into the output fidelity loss, \( L_{of} \). The rationale behind this approach, however, requires clarification. It would be beneficial for the authors to validate the use of output fidelity loss \( L_{of} \), particularly in comparison to employing cross-entropy loss with ground-truth labels.

- W2: The auto-encoding objective is inherently insufficient for the acquisition of compositional representations, as the optimization of reconstruction quality does not necessarily entail the disentanglement of features at the object or concept level [1]. The integration of functions \(\Psi\) and \(\Omega\) can certainly be characterized as an auto-encoder architecture. In the main text and appendix, the authors present further applications that employ various functions \( f \) and \( G \). Nevertheless, it would be advantageous to explore modern architectures, such as transformers for \( f \) and diffusion models for \( G \).

- Reference
- [1] Jung, Whie, et al. "Learning to Compose: Improving Object Centric Learning by Injecting Compositionality." arXiv preprint arXiv:2405.00646 (2024)

**Questions:**

Most of my major questions/concerns are listed in the Weakness sections.

---

> ### Author Response · Authors · 2024-11-21
> **Response to Reviewer gc79**
>
> Thank you for the review. We address your comments below:
>
> ### Weaknesses
>
> * **W1**: Yes, one can consider using cross-entropy loss with ground truth labels instead of "output fidelity loss". We specify the possibility to use both when describing the general architecture of unsupervised CoINs (Line 196). We decided to use the output fidelity loss following previous work (Sarkar et al., 2022). The current design also draws inspiration from knowledge distillation setting, in which a student model is trained to reproduce output of a teacher model. One additional perk of using this "output fidelity loss" during VisCoIN training is that it can also be applied for images without annotations, such as images sampled from G (further details about VisCoIN training in Appendix C.2.3). This provides additional guidance and stability to train $g$. For completeness, we include here an experiment of VisCoIN trained using a standard cross-entropy loss with ground truth labels on CUB dataset:
> Model | Acc. ($\uparrow$) | LPIPS ($\downarrow$) | FF ($\tau = 0.2$) ($\uparrow$)
> --- | --- | --- | ---
> VisCoIN - Output fidelity loss | **79.44** | **0.545** | **0.146**
> VisCoIN - Cross-entropy loss | 78.89 | 0.559 | 0.076
>
> * **W2:**
>     * We discuss in the limitations (Appendix H) the inherent limitations for disentangling concepts in the unsupervised learning. However, it is also a problem appearing in supervised learning of the concepts (with supervised CoINs, e.g., CBMs) that can suffer from concept leakage [A]. Furthermore, increasing the size of the concept dictionary (i.e. $K$, the number of concepts in $\Phi$) could help in learning more disentangled concepts at the cost of reducing conciseness of the dictionary, similarly to Sparse AutoEncoders (SAEs) in "mechanistic interpretability" [B]
>     * We are happy to report an additional first experiment with a transformer architecture. We started from a pretrained ViT-B/16 and only finetuned the classification head on CUB, to use as $f$. We take the patch embeddings of final layer as input to $\Psi$. We keep identical $\Theta, \Omega$, hyperparameters, and slightly modify $\Psi$ for reduced number of feature maps. As can be seen from the numerical results below, we achieve better accuracy thanks to a better pretrained $f$, but reconstruction and faithfulness are worse. The results could be improved by better desigining $\Psi$ and accessing more internals embeddings. However, the results certainly show that the idea can generalize to other backbone architectures. We discuss desiderata for $G$ and using diffusion models, in Appendix B. They would be interesting to explore as an extension, but despite recent positive steps towards understanding their latent space and discovering meaningful latent directions ([C], [D]), it is currently difficult to design an $\Omega$ that allows convenient latent traversal.
>  Model | Acc. f | Acc. g | LPIPS ($\downarrow$) | FF ($\tau = 0.2$) ($\uparrow$)
>  --- | --- | --- | --- | ---
>  VisCoIN - ResNet50 | 80.56 | 79.44 | **0.545** | **0.146**
>  VisCoIN - ViT-B/16 | 86.66 | 85.86 | 0.582 | 0.081
>
> [A] M. Havasi, S. Parbhoo, F. Doshi-Velez. "Addressing Leakage in Concept Bottleneck Models". NeurIPS 2022
>
> [B] E. Nelson, et al. "Toy models of superposition." arXiv preprint arXiv:2209.10652 (2022)
>
> [C] Y.H. Park et al. "Unsupervised Discovery of Semantic Latent Directions in Diffusion Models". NeurIPS 2023
>
> [D] M. Kwon et al. "Diffusion Models Already have a Semantic Latent Space". ICLR 2023

---

> > ### Comment · Reviewer_gc79 · 2024-11-25
> > **Reviewer gc79 response**
> >
> > Many thanks to the authors for providing additional experimental results.
> > I hope the updates will be reflected in the manuscript. I will keep my score.

---

### Official Review · Reviewer_BbA8 · 2024-11-10

**Soundness:** 3
**Presentation:** 3
**Contribution:** 2
**Rating:** 6
**Confidence:** 4

**Summary:**

This paper proposes a new approach for inherently interpretable models by mapping the concept space to the latent space of a pre-trained generative model. In particular, this approach focuses on using this mapping to associate semantics with the concepts. To this end, the paper proposes the use of three adapter modules along with a pre-trained classifier and generator to achieve the goal. Appropriate losses and metrics are defined to train and evaluate the method. The experiments on standard large-scale benchmark datasets shows promise in the approach.

**Strengths:**

+ The idea of using the latent space of a generative model to map implicitly learned concepts of a neural network model is interesting.
+ The methodology is simple and effective.
+ The paper is well-written with a good treatment of relevant literature.
+ The method and experiments are well-documented for reproducibility.

**Weaknesses:**

- One fundamental concern (a part of which is briefly discussed in Appendix A): considering recent concept-based models that use LLMs for semantics, it can be argued that concepts can directly be interpreted through human-understandable language semantics. How important is visualization in such a scenario? Is it possible to show through some user studies that a user necessarily requires visualization beyond just language semantics in real-world applications? Without this, the premise of this work may be weak.

- A second major concern is the limited baselines used for experimental comparison. Many baselines seem missing: Label-free CBMs, LaBo, Posthoc CBMs, Sparse CBMs. It is not clear why some of these were not considered -- it may not be difficult to adapt some of them for comparison. Further, while some of them lack "visualizability", comparing the proposed method w.r.t. these baselines on other metrics is important for completeness of understanding.

- While the appendix reports many ablation studies, I found in general a lack of depth of analysis of the results, with a propensity of very brief analysis of multiple factors. I would have preferred seeing at least some of the important analysis being carried out in depth. In fact, I found the appendix hard to parse since there were too many studies, but with too little discussion on inferences and take-aways.

- One reference that is close to this work is: "Garg et al, Advancing Ante-Hoc Explainable Models through Generative Adversarial Networks, arXiv:2401.04647, AAAI-W'24". It may be good to compare the proposed work against this paper, since they have similar objectives esp the viewability property.

**Questions:**

Please see weaknesses above. Below are some additional questions:
* Are there any restrictions of what f and G should be pre-trained on? How close should those datasets be to the one being studied? Since interpretability is the focus of this work, it would be useful to know how semantically related these must be.

---

> ### Author Response · Authors · 2024-11-21
> **Response to Reviewer BbA8**
>
> ### Weaknesses
>
> We thank the reviewer for their insightful comments. We answer pointwise below, and will add the discussions in the revised version of the paper.
>
> * **W1:** In relation to LLM/VLM based concept bottleneck models (CBMs), we present below arguments why visualizing concepts is still important:
>
>     * When considering expert or domain specific datasets like Stanford Cars (for car models classification) or the MVTec Anomaly Detection [A] (for anomaly detection of object in production lines) for instance, visualizing concepts directly on the objects is simpler *(and faster)* to understand for human operators, rather than reading a text description.
>
>     * For certain computer vision applications (eg. self-driving cars, medical imaging tasks), **visualization provides spatially localized interpretations, which is more difficult and cumbersome with text**. For instance, if a concept relating to "red light" is activated for an image, to get a thorough understanding of the model's decision, it is crucial to identify which regions and what content in those regions activates the concept.
>
>     * The LLMs/VLMs which the recent CBMs are based on (particularly CLIP) are limited when detecting concepts and image details at a finer spatial scale [B].
>
>     * As discussed in Appendix A, the current methods are prone to **generating concept descriptions not grounded in any visual information**, which also harms their interpretability. Take the following example. We have an image of a dog, and want to know which concepts lead to the classification of the image as a dog. An LLM or VLM might, for instance, introduce concepts such as "Loyal/Honest" for detection and interpretation, ie: "the object in this image is loyal, therefore it is a dog". In this situation, "loyal" should not be even generated as a concept, since it can never be visually observed.
>
>     * In the case of LLM/VLM based CBMs, there are also concerns about **faithfulness of concept detection to the text description**. This is a similar issue to concept leakage [C]. We believe that the ideas presented in our work, such as viewability, can help in identifying such issues in LLM/VLM based CBMs.
>
>     [A] P. Bergmann et al. (2021). "The MVTec anomaly detection dataset: a comprehensive real-world dataset for unsupervised anomaly detection."" IJCV.
>
>     [B] C. Gou et al. "How Well Can Vision Language Models See Image Details?". https://arxiv.org/pdf/2408.03940
>
>     [C] M. Havasi et al. "Addressing Leakage in Concept Bottleneck Models". NeurIPS 2022.
>
>
> * **W2:**  The methods mentioned, among others, are separately defined by us as "supervised CoINs" (Section 3.1), since they use concept annotations to learn $\Phi$. Although these concepts annotations can be "automatically" obtained from LLMs and VLMs, they are extracted/generated beforehand to train the underlying "CBM". On the other hand, all the baseline methods we compare to "discover" concepts in an unsupervised way without concepts annotations, which we define as "unsupervised CoINs".
>
>     A **major difference** which makes adapting or comparison with them difficult, is **their inherent lack of "decoder" model**, that prevents us from going back to the input space from the concept activations. Without a decoder, we cannot evaluate consistency, faithfulness, and reconstruction metrics, making a comparison only available for accuracy. We will add this clarification in the paper.
>
>
> * **W3:** Thanks for the remark. We'll improve the clarity by better separating the sections, discussing experiments in more depth, and add a summary of take-aways for them.
>
>
> * **W4:** Thank you for the reference. It is indeed an unsupervised CoIN that includes a GAN as a decoder model. However, it differs significantly from VisCoIN in two major aspects. First, they learn the generative model simultaneously along with other constraints. This can make the overall training challenging, since GANs are notoriously difficult to train. The training will also be more costly for large scale images and bigger generators. Second, they do not leverage the GAN for visualizing the concepts. They only use the maximum activating samples (MAS) for visualization. Thus viewability is not part of their aims. The GAN is used as a decoder with higher expressivity, with the goal of improving accuracy.
>
>     Unfortunately, we didn't find any public codebase to help reproduce and compare with this method. The experiments in the paper are only on small scale image datasets like CIFAR10 and CIFAR100.
>
> ### Questions
>
> * **Q1:** In our experiments, we finetuned both $f$ and $G$ on the datasets being studied, before training VisCoIN. We didn't study how much these restrictions can be lifted. We expect that the pretrained $f$ and $G$ should have good accuracy and generative capabilities on the dataset being studied even if they aren't trained on it. However it's an interesting future research direction to explore. Thanks for raising the point.

---

> > ### Comment · Reviewer_2hPr · 2024-11-25
> >
> > Dear Reviewer BbA8,
> >
> >  I wonder if you could share where you stand on this response? Are there any further concerns or questions that you would like to highlight before the end of the discussion period?

---

> > ### Comment · Reviewer_BbA8 · 2024-11-27
> > **Response to authors' rebuttal**
> >
> > Thank you to the authors for the detailed responses. I had a few follow-up queries/comments:
> >
> > * W1: Thank you for highlighting the need for concept visualization. A couple of further thoughts: (i) a human needs to manually intervene to understand a visualization, but a text semantic is more easily transferable to downstream applications (eg. a document or a text-to-speech converter), and (ii) ideally, showing the need for this using some user studies (e.g. providing language concepts and visualization concepts and checking if this adds more understanding) may be more ideal to showcase the usefulness. Adding this discussion to the paper may be important to position the usefulness of this work.
> > * W2: This is not convincing. Methods like Label-free CBMs do not require labeled concept data -- I believe comparison with such methods, which actually work well, may be necessary for completeness.
> > * W4: Thank you for the clarification -- I agree. It may be good to add this to related work, and clearly differentiate.
> >
> > Since the rebuttal phase so far also encouraged the authors to update the manuscript, I'd appreciate if the authors could point to the edits based on the review comments and responses. (I'd have especially been happy to see this for W3 in the rebuttal -- I found the appendix to lack insightful discussions).
> >
> > PS: I am sorry for the slow follow-up, but am happy to engage in discussion until the deadline from here.

---

> > > ### Author Response · Authors · 2024-11-27
> > >
> > > Thank you for your response to our rebuttal
> > >
> > > * **W1, W2, W3 and W4:** We have updated the paper to include the respective discussions or to address the concerns. In case there were particular details you were looking for, that lack in appendix (W3), please let us know, we'll make a note of them to update later.
> > >
> > > * **W2:** There is some misunderstanding here. We state in our rebuttal that for LLM/VLM based CBMs, "concept annotations" can be automatically obtained using LLMs/VLMs, so we already agree there is no need to **manually** annotate concepts. However, methodologically, instead of using human annotations, they use CLIP-similarities of text descriptions with image to learn the bottleneck layer. Our statement about "using concept annotations" is from a methodological perspective to highlight this common paradigm.
> > >
> > >     Nevertheless, our **main point** about the comparison still remains as is. These models **do not have any decoder**. There is no way to conduct any of the main evaluation about quality of interpretation (reconstruction, faithfulness, consistency) without a decoder, because we need to approximate the input from concept activations to do any of these. All unsupervised CoINs have a decoder and none of the CBMs do, as a result of their modelling and training methodological differences. Including a decoding branch into CBMs is not a trivial modification. We updated the paper to make this distinction more explicit.
> > >
> > >     If you wish we can report accuracy of all models on CUB-200, since it is the only metric we can compare on, and all models have been evaluated on this dataset. However, we don't feel it adds a meaningful comparison.

---

> ### Comment · Reviewer_BbA8 · 2024-12-01
>
> I thank the authors for the responses, and have increased the score to 6.
>
> While I understand the manuscript cannot be updated any further, my point about W2 was that -- if you adapted (eg. attached a simple decoder to) an existing method such as Label-free CBMs, would that outperform such an approach? I could not follow why this is non-trivial. I'd appreciate if a discussion on this can be added, but I'd understand otherwise too.

---

> > ### Author Response · Authors · 2024-12-02
> >
> > Thank you for the positive update and clarification about your point!
> >
> > A high quality reconstruction would very likely still require a complex decoder. Otherwise, viewability is much harder to enforce and can result in poor LPIPS/FID metrics. We consider this setup non-trivial because in contrast to unsupervised CoINs, the concept representation in CBMs is directly constrained by the concept annotations or CLIP similarity scores. Whether the decoder (preferably pretrained) is still able to map such a constrained $\Phi$ to its latent space and generate high-quality reconstructions remains a challenging research question.
> >
> > Still we agree this presents a valuable future research direction to explore. This setup is one pathway to progress upon the final point we discussed for W1. We will be happy to add this discussion in Appendix A.

---

### Author Response · Authors · 2024-11-27
**Summary of revision (global comment)**

We sincerely thank **all the reviewers** for their reviews, suggestions, and rebuttal acknowledgements. Your engagement overall has really helped us improve our work.

We have updated the paper with a revised version taking into account all comments until now. It incorporates the following changes:

* Main paper (Reviewer BbA8): Added Garg et al. reference and discussion about it in Sec 3.1. We also added more detailed discussion about LLM/VLM based CBMs in Sec 3.1 and Appendix A (with a reference to it in Sec 2).
* Main paper (Reviewer 9qiE): Reference of architectural details in appendix added in Sec 3.2.
* Main paper (Reviewers Lumc, 2hPr): Details about "naming" the concepts and red boxes in figure captions in Sec 4.2
* Appendix general changes (Reviewer BbA8): We added an introduction to Appendix describing the broad organization. The various sections of appendix along with their figures and tables are better separated. We added more discussion to ablation studies (Appendix F), and some other places in appendix, where we felt depth and details were lacking.
* Appendix E (Reviewers Lumc, 2hPr): All experiments evaluating the model further are now clubbed together in Appendix E: "Additional Evaluation". This includes AUC-Faithfulness metric with greater resolution, sparsity evaluation, the additional consistency evaluation, and top-N activation filter.
* ViT experiment (Reviewers gc79, 2hPr) is integrated in Appendix D.

The edited or newly added parts are indicated in blue. We will be happy to discuss further in case of any doubts.

---

### Meta-Review · Area_Chair_qTtg · 2024-12-19

**Metareview:**

This paper  proposes a concept translator for inherently interpretable models. This translator uses pre-trained generative models, and maps concepts into the learned latent space of this pre-trained generative model.

Important concerns, such as relation with CBM models, the rationale of introducing an additional pre-trained classifier f, the details of how reconstruction is performed, have been raised by reviewers and addressed by authors.

After rebuttal, the paper received unanimous accept recommendations from all 5 reviewers (6, 6, 8, 6, 6). I am on board with them.

**Additional Comments On Reviewer Discussion:**

Some open discussion about visualization v.s. language semantics have been discussed. While some concerns have been thoroughly discussed, the concerns from reviewer Lumc still exist.

While I recommend accepting this paper, I would encourage the author to address them in the final version.

---

### Decision · Program_Chairs · 2025-01-22

Accept (Poster)